# DNALI1 interacts with the MEIG1/PACRG complex within the manchette and is required for proper sperm flagellum assembly in mice

Yi Tian Yap[1†], Wei Li[1†], Qian Huang[1,2†], Qi Zhou[1,2], David Zhang[3], Yi Sheng[4], Ljljiana Mladenovic-Lucas[5], Siu-Pok Yee[6], Kyle E Orwig[4], James G Granneman[5], David C Williams Jr[7], Rex A Hess[8], Aminata Toure[9], Zhibing Zhang[1,10]*

[1]Department of Physiology, Wayne State University School of Medicine, Detroit, United States; [2]Department of Occupational and Environmental Medicine, School of Public Health, Wuhan University of Science and Technology, Wuhan, China; [3]College of William and Mary, Williamsburg, United States; [4]Molecular Genetics and Developmental Biology Graduate Program, Department of Obstetrics, Gynecology and Reproductive Sciences, Magee-Womens Research Institute, University of Pittsburgh School of Medicine, Pittsburgh, United States; [5]Center for Molecular Medicine and Genetics, Wayne State University School of Medicine, Detroit, United States; [6]Department of Cell Biology, University of Connecticut Health Center, Farmington, United States; [7]Department of Pathology and Laboratory Medicine, University of North Carolina, Chapel Hill, United States; [8]Department of Comparative Biosciences, College of Veterinary Medicine, University of Illinois, Urbana, United States; [9]University Grenoble Alpes, Inserm U 1209, CNRS UMR 5309, Team Physiology and Pathophysiology of Sperm cells, Institute for Advanced Biosciences, Grenoble, France; [10]Department of Obstetrics & Gynecology, Wayne State University, Detroit, United States

*For correspondence:
gn6075@wayne.edu

†These authors contributed equally to this work

**Abstract** The manchette is a transient and unique structure present in elongating spermatids and required for proper differentiation of the germ cells during spermatogenesis. Previous work indicated that the MEIG1/PACRG complex locates in the manchette and is involved in the transport of cargos, such as SPAG16L, to build the sperm flagellum. Here, using co-immunoprecipitation and pull-down approaches in various cell systems, we established that DNALI1, an axonemal component originally cloned from *Chlamydomonas reinhardtii*, recruits and stabilizes PACRG and we confirm in vivo, the co-localization of DNALI1 and PACRG in the manchette by immunofluorescence of elongating murine spermatids. We next generated mice with a specific deficiency of DNALI1 in male germ cells, and observed a dramatic reduction of the sperm cells, which results in male infertility. In addition, we observed that the majority of the sperm cells exhibited abnormal morphology including misshapen heads, bent tails, enlarged midpiece, discontinuous accessory structure, emphasizing the importance of DNALI1 in sperm differentiation. Examination of testis histology confirmed impaired spermiogenesis in the mutant mice. Importantly, while testicular levels of MEIG1, PACRG, and SPAG16L proteins were unchanged in the *Dnali1* mutant mice, their localization within the manchette was greatly affected, indicating that DNALI1 is required for the formation of the MEIG1/PACRG complex within the manchette. Interestingly, in contrast to MEIG1 and PACRG-deficient mice, the DNALI1-deficient mice also showed impaired sperm spermiation/individualization, suggesting additional functions beyond its involvement in

the manchette structure. Overall, our work identifies DNALI1 as a protein required for sperm development.

## Editor's evaluation

This study provides new insights into the role of DNALI1, an axonemal dynein component, in the machete, a unique transient structure in elongating spermatids during normal spermiogenesis of male germ cell differentiation. The authors provide convincing evidence that DNALI1 is associated with the MEIG1/PACRG complex, which is part of intra-manchette transport system, and is required for proper flagellar assembly and male fertility.

## Introduction

Motile cilia are microtubule-based organelles with a core '9+2' axonemal structure (*Dutcher, 1995*). They generate fluid flow by their beating in various epithelia and also enable sperm cell progression, which overall confers vital functions in eukaryotes (*Neesen et al., 2002*; *Ibañez-Tallon et al., 2002*). In humans, motile cilia dysfunction results in multiple syndromes, including bronchiectasis, impaired mucociliary clearance, chronic cough, sinusitis, and male infertility, which are described as immotile cilia syndrome, as well as *primary ciliary dyskinesia* (PCD) (*Leigh et al., 2009*; *Noone et al., 2004*). Ciliary beat is driven by dynein motor protein complexes, which are subdivided into the inner and outer arms of axonemal ultrastructure. These dynein arms are anchored to the peripheral axonemal microtubules and are the motor proteins that power the sliding of adjacent microtubule doublets relative to one another through ATP hydrolysis. The heavy, medium, and light chains constitute the dynein motors, and each chain has a different molecular weight and provides an important function (*Gibbons, 1995*; *King, 2000*). Hence, in mammals the loss of dynein function leads to PCD with ciliary impairment with hydrocephalus, body axis asymmetry, and male infertility (*Ibañez-Tallon et al., 2002*; *Neesen et al., 2001*; *Supp et al., 1997*).

Human axonemal dynein light intermediate polypeptide 1 (*DNALI1*) gene is the homolog of the *Chlamydomonas* inner dynein arm (IDA) gene *p28*, an important component of the ciliated dynein arm, which is mainly responsible for cilium movement (*LeDizet and Piperno, 1995*; *Kastury et al., 1997*). The molecular analysis of *DNALI1* (human *hp28*) gene was revealed by Dr. Shalender Bhasin's laboratory more than 20 years ago. The gene was localized on chromosome 1 region p35.1, and *DNALI1* transcripts were detected in several ciliary structures, including sperm flagella, suggesting that this gene could be a good target for patients suffering from immotile cilia syndrome (*Kastury et al., 1997*). In addition to PCD, *DNALI1* was also related to a variety of diseases, including fronto-temporal lobar degeneration, diploid breast carcinoma, osteosarcoma, allergic rhinitis, nasopharyngeal carcinoma, and Alzheimer's disease (*Mishra et al., 2007*; *Loges et al., 2009*; *Parris et al., 2010*; *Tian et al., 2018*; *Peng et al., 2018*; *Ye et al., 2019*; *Piras et al., 2019*), although the precise mechanisms are unclear. But to date, the role of *DNALI1* in male germ cells and in human male reproduction was never investigated.

Recently, *Dnali1* was identified as a sex-related gene involved in spermatogenesis of several fishes, including the *Odontobutis potamophila*, the Ussuri catfish *Pseudobagrus ussuriensis*, and the olive flounder *Paralichthys olivaceus* (*Wang et al., 2019*; *Pan et al., 2021*; *Wang et al., 2021*). The expression of *Dnali1* showed sexual dimorphism with predominant expression in the testis. This suggested that the fish *Dnali1* might play an important role in the testis, especially in the period of spermatogenesis (*Wang et al., 2021*). Sajid et al. cloned the murine *Dnali1* with significant similarity to the *p28* gene of *Chlamydomonas reinharditii* and to *DNALI1* (*hp28*) gene (*Rashid et al., 2006*). The murine *Dnali1* gene is localized on chromosome 4 and consists of six exons. It has two transcripts and is expressed in several tissues but the strongest expression was observed in testis. During the first wave of spermatogenesis, both *Dnali1* mRNA and protein were dramatically increased during the spermiogenesis phase, which corresponds to spermatid cell differentiation. Immunofluorescence studies demonstrated that DNALI1 was detected in spermatocytes and abundant in round and elongated spermatids. Moreover, the DNALI1 protein was localized in the flagella of mature sperm cells (*Rashid et al., 2006*), indicating that *Dnali1* may play an essential role in murine sperm formation and male fertility.

Sperm production and formation is a complex process consisting of mitosis, meiosis, and finally spermiogenesis (*Hecht, 1998*). During spermiogenesis, germ cells undergo dramatic changes with the formation of sperm-unique structures, including the flagellum (*Pleuger et al., 2020*). Many genes have been described to precisely regulate these morphologic changes (*Yan, 2009*). We previously discovered that mouse meiosis-expressed gene 1 (MEIG1) and Parkin co-regulated gene (PACRG) form a protein complex in the manchette, a transient microtubule structure localized in the head compartment of sperm cells during differentiation. It was demonstrated that this protein complex controls sperm flagellum formation, with PACRG being an upstream molecule of MEIG1 (*Zhang et al., 2009*; *Li et al., 2015*; *Li et al., 2016*; *Li et al., 2021*). The MEIG1/PACRG complex was shown to function as part of the cargo transport system, called intramanchette transport (IMT) system (*Kierszenbaum, 2002*), which transports sperm flagellar proteins, including SPAG16, along the microtubule or F-actin-based transport system to the basal bodies, the template of cilia (*Marshall, 2008*), for sperm tail assembly. However, how the MEIG1/PACRG complex associates with the axonemal motor system in the manchette for cargo transport remained unclear. From the basal bodies, cilia/flagella are formed through a conserved mechanism, the intraflagellar transport (IFT). IFT is a bidirectional process. Through antegrade transport, the cargo proteins are transported from the basal bodies to the tips of cilia/flagella and by the retrograde transport it is possible for the turn-over products to be transported back to the cell body for recycling (*Rosenbaum and Witman, 2002*).

Using mouse PACRG as bait for a yeast two-hybrid screen, DNALI1 was identified as a protein binding partner. To investigate the function of DNALI1, a *Dnali1* conditional knockout (cKO) mouse model was generated using a conditional by inversion (COIN) strategy (*Economides et al., 2013*). This *Dnali1* mouse line, *Dnali1^{COIN/COIN}* was crossed with *Stra8*-iCre transgenic mice, to specifically inactivate the *Dnali1* gene in the male germ cells. Reported here, the male germ cell-specific knockout *Dnali1* (*Dnali1^{COIN/COIN}*; *Stra8-iCre^{+/-}*; hereafter *Dnali1* cKO) mice have significantly reduced sperm number and are infertile. In addition, the majority of sperm exhibited abnormal morphology, strongly indicating the importance of *Dnali1* in the development of sperm flagella. Importantly, in the *Dnali1* cKO spermatid germ cells, MEIG1 and SPAG16L were not present in the manchette, indicating that DNALI1 is required for MEIG1/PACRG complex localization and for transport of cargo proteins to assemble the sperm flagellum. Some unique phenotypes discovered in the *Dnali1* mutant mice but not in the MEIG1 and PACRG-deficient mice suggest additional roles of the gene in sperm cell differentiation.

**Table 1.** List of putative PACRG binding proteins selected under stringent conditions. The full-length PACRG coding sequence was cloned into pGBKT7, which was used to screen a Mate & Plate Library-Universal Mouse (Normalized) (Clontech, Cat No: 630482) according to the manufacturer's instructions. The yeasts were grown on plates lacking four amino acids (Ade-Leu-His-Trp). DNALI1 was found to be one of the putative PACRG binding proteins.

| Name | NCBI number | Frequency |
|---|---|---|
| Meig1 | NM_008579 | 119 |
| Dnali1 | NM_175223 | 6 |
| Pramel42 | NM_001243938 | 3 |
| Musculus protein phosphatase 1A | BC008595 | 2 |
| Acad11 | NM_175324 | 2 |
| Ppm1a | NM_008910 | 1 |
| L2hgdh | NM_145443 | 1 |
| Tmem225 | NM_029379 | 1 |
| Tinag | NM_012033 | 1 |
| Spag6l | NM_015773 | 1 |
| Emp2 | NM_007929 | 1 |

## Results

### DNALI1 is a binding partner of PACRG

PACRG, a major spermatogenesis regulator, was used as a bait for yeast two-hybrid screening. The sequencing results of positive clones revealed that MEIG1 was the major binding partner as previously described (*Zhang et al., 2009*). In addition, DNALI1 encoding clones were identified multiple times (*Table 1*). A direct yeast two-hybrid assay was conducted to confirm the interaction between PACRG and DNALI1. Like the positive control, the yeast co-transformed with the two plasmids expressing

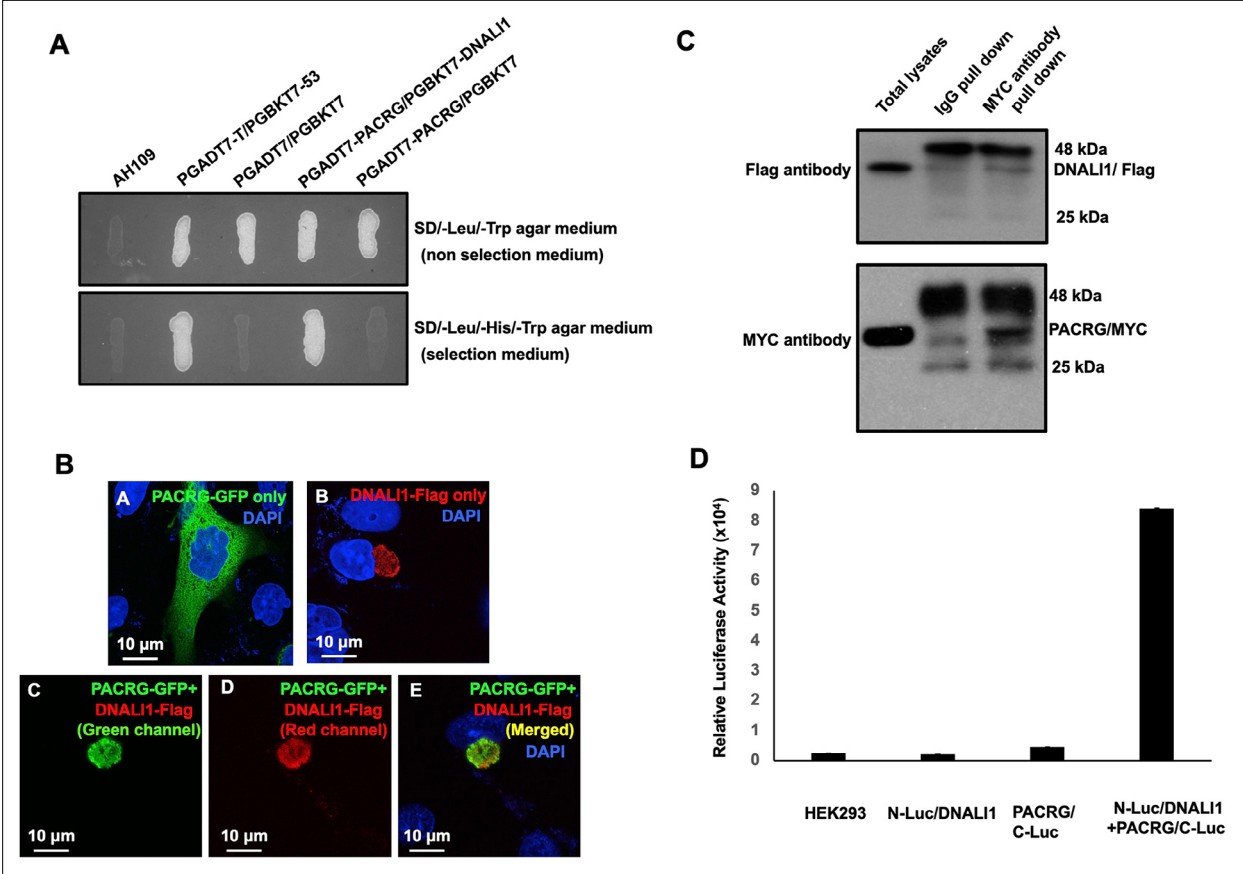

**Figure 1.** DNALI1 associates with PACRG, a major spermatogenesis regulator. (**A**) Direct yeast two-hybrid assay to examine the interaction between PACRG and DNALI1. Pairs of indicated plasmids were co-transformed into AH109 yeast, and the transformed yeast were grown on either selection plates (lacking leucine, histidine, and tryptophan) or non-selection plates (lacking leucine and tryptophan). Notice that all the yeast except AH109 grew on the non-selection plate. Yeast expressing PACRG/DNALI1 and P53/large T antigen pairs grew on selection plate. (**B**) DNALI1 co-localizes with PACRG in Chinese hamster ovarian (CHO) cells. When expressed alone, PACRG/GFP was present in the cytoplasm, and DNALI1/FLAG as a granule located closed to the nucleus. When the two proteins were co-expressed, DNALI1/FLAG recruited the PACRG/GFP to the granule structure. Images were taken with a laser scanning confocal microscopy (Zeiss LSM 700, Virginia Commonwealth University). (**C**) Co-immunoprecipitation of DNALI1/FLAG with PACRG/Myc. COS-1 cells were transfected with plasmids to co-express DNALI1/FLAG and PACRG /Myc. The cell lysate was immunoprecipitated with anti-MYC antibody and then analyzed by western blotting with anti-MYC and anti-FLAG antibodies. The cell lysate immunoprecipitated with a mouse normal IgG was used as a control. The anti-MYC antibody pulled down both PACRG/MYC and DNALI1/FLAG. (**D**) Interaction of PACRG with DNALI1 in HEK293 cells as determined by *Gaussia princeps* luciferase complementation assay. HEK293 cells were transfected with the indicated plasmids, and luciferase activity was evaluated 24 hr after transfection. The cells expressing both N-Luc/DNALI1 and PACRG/C-Luc reconstituted activity.

The online version of this article includes the following source data and figure supplement(s) for figure 1:

**Source data 1.** Co-immunoprecipitation of DNALI1/FLAG with PACRG/Myc.

**Figure supplement 1.** DNALI1 does not bind to MEIG1.

**Figure supplement 1—source data 1.** Co-immunoprecipitation of DNALI1/FLAG with MEIG1.

DNALI1 and PACRG grew on the selection medium (*Figure 1A*), indicating that the two proteins interact in yeast. To further examine interaction of the two proteins, Chinese hamster ovarian (CHO) cells were transfected with the plasmids expressing the two proteins. When the CHO cells expressed PACRG/GFP only, the protein was present in the cytoplasm (*Figure 1Ba*) while DNALI1/FLAG was located on one side of the nucleus as a large granule (*Figure 1Bb*). When CHO cells expressed both proteins, DNALI1/FLAG recruited the PACRG/GFP to the granule structure (*Figure 1Bc–e*). In addition, we transfected COS-1 cells with DNALI1/FLAG and PACRG/Myc expression plasmids and conducted a co-immunoprecipitation assay. The MYC antibody pulled down both the Myc-tagged 28 kDa PACRG and Flag-tagged 34 kDa DNALI1 proteins, suggesting an interaction between these proteins (*Figure 1C*). As a control experiment, co-immunoprecipitation assay was conducted using

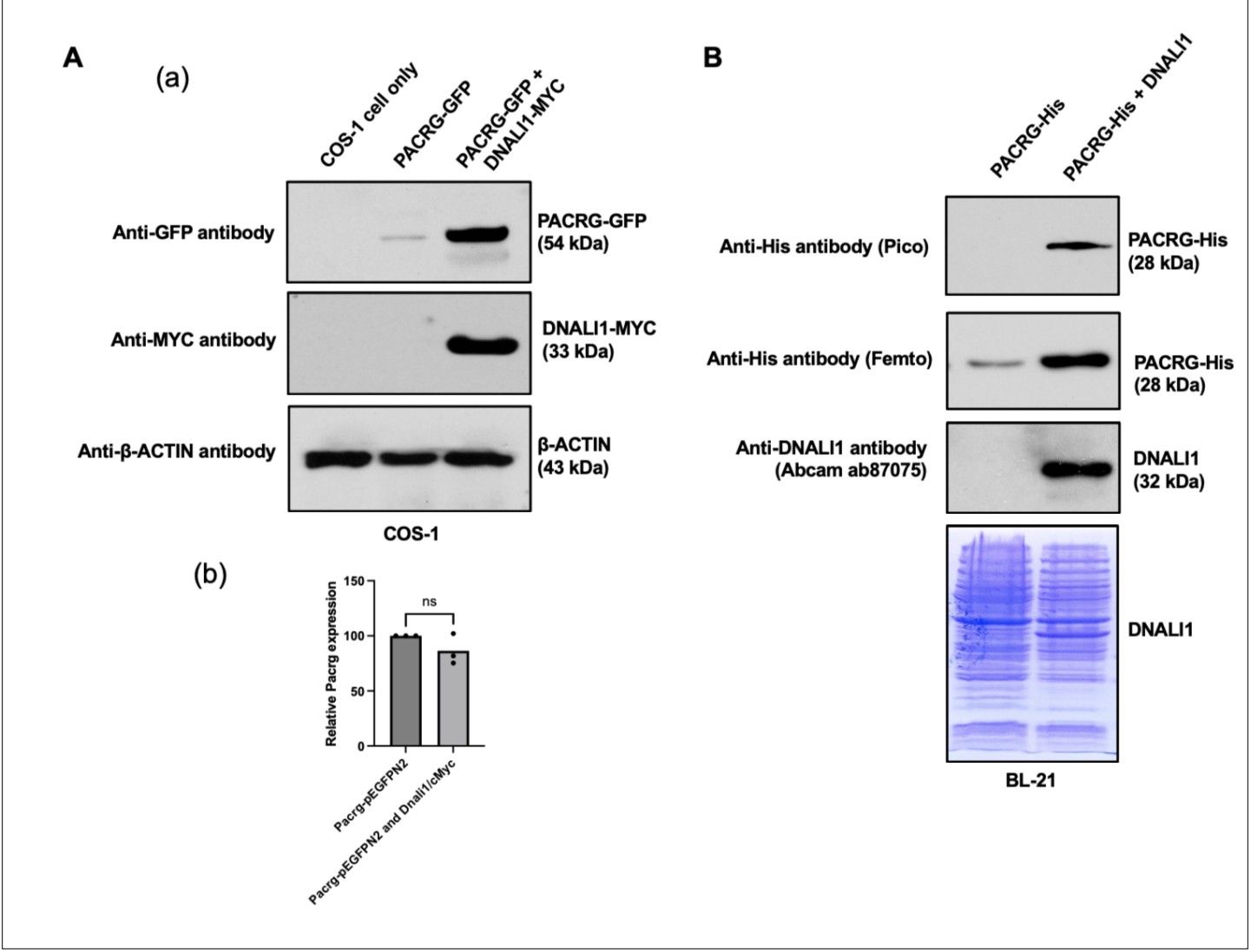

**Figure 2.** DNALI11 stabilizes PACRG in mammalian cells and bacteria. (**A**) DNALI1 stabilizes PACRG in COS-1 cells. (**a**) Mouse PACRG-GFP expression is increased when DNALI1 is co-expressed in transfected COS-1 cells in transient expression experiment. (**b**) *Pacrg* mRNA is similar in COS-1 cells transfected with PACRG-GFP alone and PACRG-GFP and DNALI1-MYC. (**B**) DNALI1 stabilizes PACRG in bacteria. Notice that PACRG was only detectable by the high sensitivity Femto system in western blot analysis when the bacteria were transformed with PACRG/pCDFDuet-1 plasmid. However, when the bacteria were transformed with PACRG/DNALI1/pCDFDuet-1 plasmid to express DNALI1 protein, PACRG was also detectable by less sensitive Pico system. Total protein lysate is shown by Coomassie stain of SDS-page gel.

The online version of this article includes the following source data for figure 2:

**Source data 1.** DNALI11 stabilizes PACRG in mammalian cells and bacteria.

COS-1 cells co-transfected with plasmids expressing DNALI1/FLAG and MEIG1. The anti-Flag antibody pulled down DNALI1/FLAG but not MEIG1 (*Figure 1—figure supplement 1*). Finally, interaction between DNALI1 and PACRG was examined by luciferase complementation assay. We observed that HEK cells co-expressing PACRG/C-Luc and N-Luc/DNALI1 showed robust luciferase activity, while the cells expressing either PACRG/C-Luc or N-Luc/DNALI1 only showed baseline activity (*Figure 1D*).

## DNALI1 stabilizes PACRG in mammalian cells and bacteria

Functional association between PACRG and DNALI1 was further supported by the fact that DNALI1 stabilized PACRG in mammalian cells and bacteria. PACRG alone was not stable in the transfected COS-1 cells with very low protein amount detected (*Li et al., 2015*). Co-expression of DNALI1 dramatically increased PACRG level (*Figure 2Aa*). The effect was not due to the increase of *Pacrg*

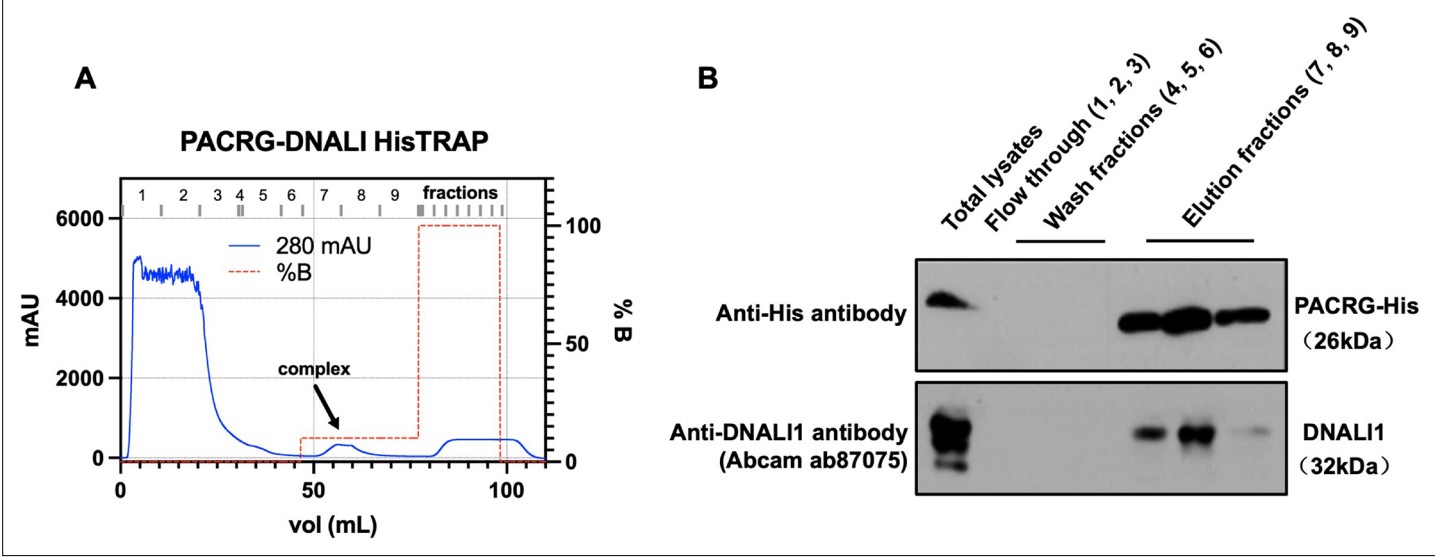

**Figure 3.** Co-purification of DNALI1 with His-tagged PACRG from bacteria lysates expressing the two proteins. (**A**) His-tagged PACRG and DNALI1 were co-expressed in BL21 bacteria and His-tagged PACRG was purified from the bacteria lysates by nickel affinity chromatography. The protein complex eluted in the low imidazole wash (arrow). (**B**) The presence of the His-tagged PACRG and non-tagged DNALI1 in the fractions were examined by western blot analysis. Notice that the DNALI1 protein was present in the same fractions of His-tagged PACRG, indicating that the two proteins associate in the bacteria lysates.

The online version of this article includes the following source data for figure 3:

**Source data 1.** Co-purification of DNALI1 with His-tagged PACRG from bacteria lysates expressing the two proteins.

mRNA level. There was no significant difference in the *Pacrg* level when the cells were transfected with PACRG-GFP alone and co-transfected with PACRG-GFP and DNALI1-MYC (*Figure 2Ab*). In BL21 bacteria, no PACRG protein was detectable when the less sensitive Pico system was used for western blot analysis after the bacteria were transformed with PACRG/pCDFDuet-1 plasmid and induced by IPTG. However, the PACRG protein was expressed as it was detectable when the higher sensitivity Femto system was used for western blot analysis. When the BL21 bacteria were transformed by PACRG/DNALI1/pCDFDuet-1 plasmid so that both PACRG and DNALI1 were expressed, the PACRG was easily detectable with the Pico system (*Figure 2B*), indicating that wild-type DNALI1 also stabilizes PACRG protein in bacteria.

## DNALI1 is co-purified with His-tag PACRG when the two proteins are co-expressed in BL21 bacteria

In order to examine the association between DNALI1 and PACRG in bacteria, we co-expressed DNALI1 and His-tagged PACRG in BL21 cells, and purified the His-tagged PACRG protein by nickel affinity chromatography. Detection of His-tagged PACRG and DNALI1 proteins in the eluted fractions was examined by western blot using specific antibodies. In line with the above interaction data, DNALI1 protein was co-purified with the His-tagged PACRG protein in the same fractions (*Figure 3*).

## Localization of DNALI1 in the manchette is not dependent on PACRG

To determine if DNALI1 and PACRG interacts in vivo, we performed co-immunoprecipitation experiments in mouse testis protein extracts. Using anti-PACRG antibody, we searched for the presence of DNALI1 protein in the precipitated complex by performing western blot using specific antibody against DNALI1. The results showed that anti-PACRG antibody specifically pulled down DNALI1 (*Figure 4A*). Next, DNALI1 localization in germ cells was examined by immunofluorescence staining. In elongating spermatids, DNALI1 co-localized with the α-tubulin, a manchette marker. Double staining with an anti-DNALI1 polyclonal antibody and an anti-PACRG monoclonal antibody revealed that the two proteins also co-localized (*Figure 4B*). In *Pacrg* mutant mice, DNALI1 was still present in the

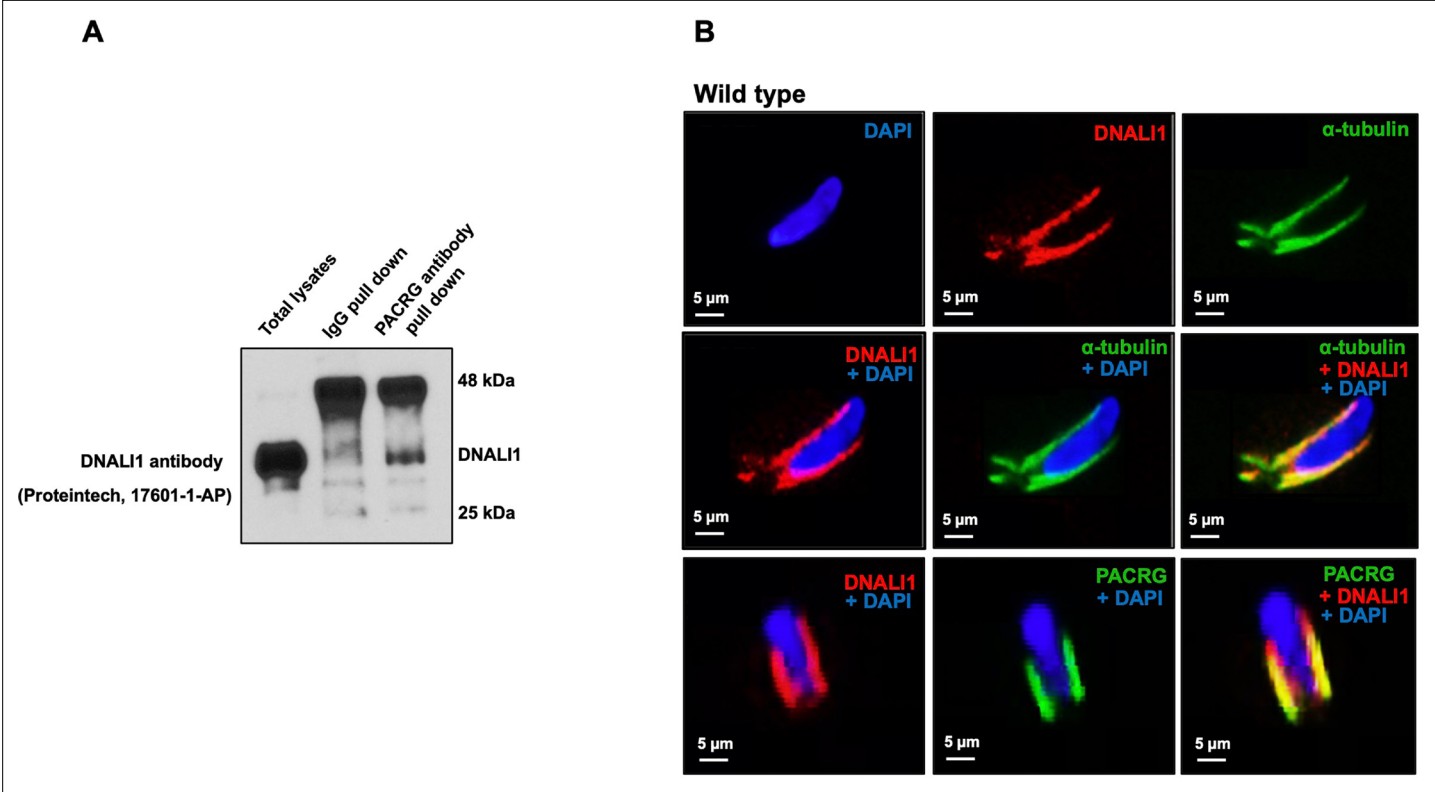

**Figure 4.** Localization of DNALI1 in male germ cells of wild-type mice. (**A**) Co-immunoprecipitation assay. Mouse testis extract was pulled down using an anti-PACRG antibody, and western blot was conducted using an anti-DNALI1 antibody. Notice that DNALI1 was co-pulled down by the anti-PACRG antibody. (**B**) Localization of DNALI1 in isolated germ cells was examined by immunofluorescence staining. DNALI1 protein localizes with α-tubulin, a manchette marker in elongating spermatids (top and middle panels). PACRG also co-localized with DNALI1 in the elongating spermatids (bottom panel). DNALI1 seems to be closer to the nuclear membrane, and PACRG is on the surface of DNALI1. Images were taken with a laser scanning confocal microscopy (Zeiss LSM 700), Virginia Commonwealth University.

The online version of this article includes the following source data and figure supplement(s) for figure 4:

**Source data 1.** Co-immunoprecipitation assay shows DNALI1 was co-pulled down by the anti-PACRG antibody.

**Figure supplement 1.** The localization of DNALI1 in the testis seminiferous tubule of a wild-type mouse (**A**) and a *Pacrg* mutant mouse (**B**).

manchette, indicating that the localization of DNALI1 in the manchette is not dependent on PACRG (*Figure 4—figure supplement 1*).

## Inactivation of mouse *Dnali1* gene resulted in male infertility associated with significantly reduced sperm number, motility, and an increase in abnormal sperm

To explore the role of DNALI1 in vivo, three ES cell clones with conditional potential were previously purchased from HelmholtzZentrum munchen (ES cell clone number: EPD0750_5_E03, order number: MAE-4178) to be injected into blastocysts in order to generate *Dnali1* cKO mice. Although chimeric mice were obtained, germline transmission was not successful as evaluated by genotyping at least 200 pups. As a result, we used the COIN strategy to generate *Dnali1* cKO mice. To this end, we used CRISPR/Cas9 to insert a COIN cassette in intron 1 of *Dnali1* to generate the cKO mouse line (see supplemental for details) (*Figure 5—figure supplement 1a*). *Dnali1^{COIN/COIN}* mice were crossed with *Stra8-iCre* mice to obtain *Dnali1^{COIN/COIN}; Stra8-iCre^{+/-}* mice (*Figure 5—figure supplement 1b*; hereafter *Dnali1* cKO) with *Dnali1* gene specifically disrupted in male germ cells (*Dnali1* cKO). Genotyping using specific primer sets indicated that we have obtained homozygous mutant *Dnali1* cKO mice (*Figure 5—figure supplement 1c*). Subsequent examination of total testicular DNALI1 protein expression by western blot revealed that DNALI1 was almost absent in the cKO mice (*Figure 5A*, *Figure 5—figure supplement 2*). Homozygous *Dnali1* cKO mice did not show any gross abnormalities,

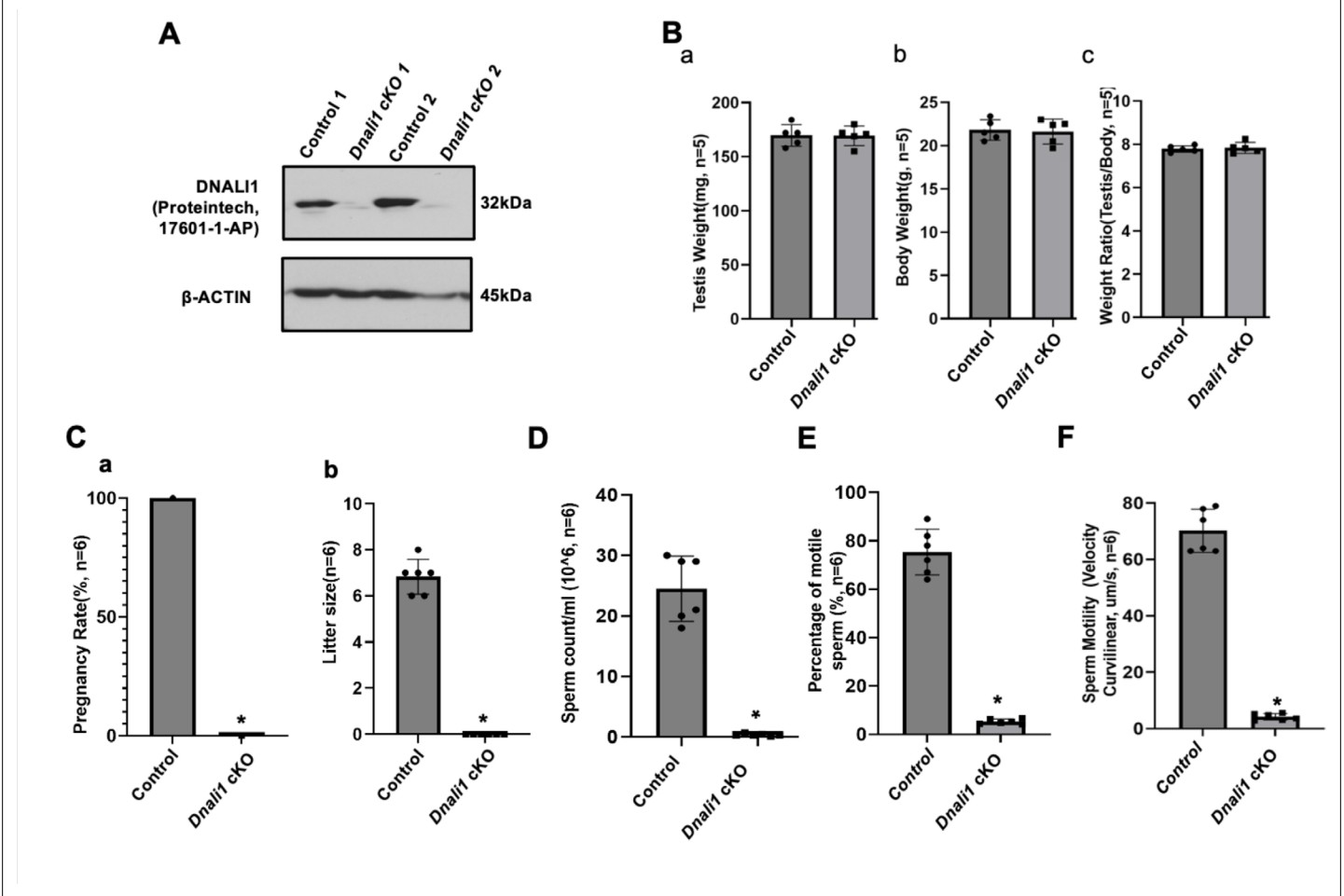

**Figure 5.** Male germ cell-specific *Dnali1* knockout, *Dnali1* conditional knockout (cKO) mice were infertile associated with significantly reduced sperm number and motility. (**A**) Representative western blot result showing that DNALI1 protein was almost absent in the testis of *Dnali1* cKO mice. (**B**) Testis weight (**a**), body weight (**b**) and ratio of testis/body weight (**c**) of 4-month-old control and *Dnali1* cKO mice. There was no significant difference between the control and the *Dnali1* cKO mice. n=5. (**C**) Male fertility of control and *Dnali1* cKO mice. Six controls and six *Dnali1* cKO mice were examined. Pregnancy (**a**) and litter size (**b**) were recorded for each mating. Notice that all mutant males were infertile (n=6). (**D**) Sperm count was significantly reduced in *Dnali1* cKO mice (n=6). (**E**) Percentage of motile sperm in the control and *Dnali1* cKO mice (n=6). (**F**) Sperm motility was significantly reduced in *Dnali1* cKO mice (n=6). Statistically significant differences: *$p < 0.05$.

The online version of this article includes the following source data and figure supplement(s) for figure 5:

**Source data 1.** Western blot result showing that DNALI1 protein was almost absent in the testis of *Dnali1* conditional knockout (cKO) mice.

**Figure supplement 1.** Generation of the conditional *Dnali1* knockout (KO) mice.

**Figure supplement 1—source data 1.** Representative PCR results showing mice with different genotypes.

**Figure supplement 2.** Examination of DNALI1 in testicular sections of the control and *Dnali1* conditional knockout (cKO) mice.

**Figure supplement 3.** Morphological examination of epididymal sperm by light microscopy at low magnification.

and their body weight and testis weight were comparable with control mice (*Figure 5B*). To test the fertility of these *Dnali1* cKO mice, 2- to 3-month-old controls and homozygous *Dnali1* cKO males were bred with 2- to 3-month-old wild-type females for 2 months. All the wild-type and heterozygous *Dnali1* cKO males were fertile and litter sizes were normal. In contrast, all homozygous *Dnali1* cKO males examined were infertile (*Figure 5C*). Examination of epididymal sperm number, motility, and morphology from the control and homozygous *Dnali1* cKO mice revealed that sperm counts and motility were dramatically reduced in the *Dnali1* cKO mice (*Figure 5D, E and F* and *Figure 5—figure supplement 3*, *Video 1* and *Video 2*). Sperm from the control mice showed normal morphology (*Figure 6A*). Multiple sperm abnormalities were observed in the mutant mice, including head defects,

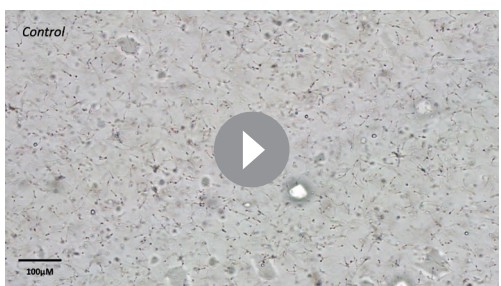

**Video 1.** Representative movie from a control mouse. Note that most sperm are motile and display vigorous flagellar activity and progressive long-track forward movement.
https://elifesciences.org/articles/79620/figures#video1

uneven thickness of tails, and sperm bundles lacking separation (also called individualization or disengagement) (*Figure 6B–F*, *Figure 6—figure supplement 1*).

## Generation of *Dnali1* conditional knockout mice

To generate *Dnali1* conditional knockout mice, we used a conditional by inversion (COIN) strategy to which an inverted gene trap cassette was inserted into intron 1 of *Dnali1* using CRISPR-mediated gene editing in C57BL6 one-cell embryos. CRISPR cleavage site (5- TGG TAG TCA AGT CCA AGA AG-3 ') within intron 1 was identified by CRISPOR (http://crispor.tefor.net). The donor vector with *Dnali1* 5' and 3' homology-arms of 780 and 695 bp, respectively, were prepared by PCR using primers: forward: 5'-GCG GCC GCT GTT CCT TGC GCG TTA TTA GTA G-3' (Dli1 5AF) and reverse 5'-GGA TCC TGG ACT TGA CTA CCA CAA ACC CCA TC (Dli1 5AR) for the 5'arm and forward: 5'- GAA TTC GGT CTG GTA GTG ACT TAC CTT CGT C-3' (Dli1 3AF) and reverse 5'-GTC GAC AAG ATG GAG GGA CCA AAG AAT GG-3' (Dli1 3AR) for the 3'arm. The COIN cassette, which contains a rabbit β-globin splice acceptor (SA) followed by eGFP that is in-frame with exon 1 coding sequence, and bovine growth hormone polyadenylation signal sequence (pA), is flanked by two mutant Lox sites, Lox66 and Lox71 in opposite direction, and was placed in between the 5' and 3' homology-arms in reverse transcriptional orientation. *Dnali1COIN/+* founders were identified by long-range PCR using primers, forward: 5'-CCT GTG GGA AAG CTA ACC CAG C-3'(DliScF5), and reverse: 5'- GCT GGG GAT GCG GTG GGC TC –3' (BGHpAr) to amplify a fragment of 844 bp for the 5'end and primers, forward: 5'-CAG CTC CTC GCC CTT GCT CAC C-3' (eGFPrf) and reverse: 5'-CCA GGC TCT CTA TGA GGA CTC-3' (Dli1 ScR3) to amplify a fragment of 794 bp specific for the 3'end. Both PCR products derived from the 5' and 3'end were sequenced to confirm their identities.

The COIN cassette containing eGFP is placed in antisense orientation with respect to *Dnali1* and it will be spliced out in the *Dnali1* transcript. However, in the presence of Cre, Cre will mediate inversion of genomic sequence flanked by Lox66 and Lox71 and convert them into LoxP and Lox72, respectively. Since Cre does

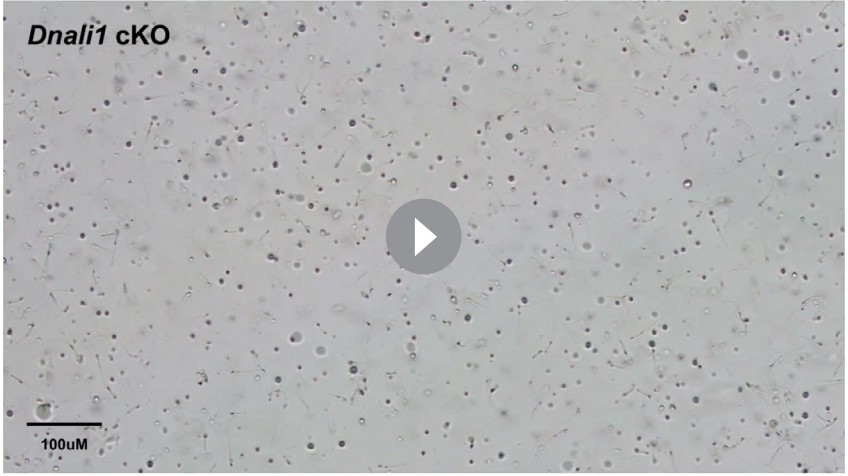

**Video 2.** Representative movies from a *Dnali1* cKO mouse. Notice that there are fewer sperm compared with the control mice in the same dilution, and almost all sperm are immotile.
https://elifesciences.org/articles/79620/figures#video2

not bind Lox72, the inversion only occurs once. The inversion will place the splice acceptor and eGFP into the same transcriptional orientation of *Dnali1* and generate a Dnali1-eGFP fusion transcript. The bGH polyadenylation sequence will terminate further transcription of *Dnali1* downstream exons leading to the lack of a *Dnali1* functional protein in *Dnali1^COIN* allele.

## Spermatogenesis was affected in the *Dnali1* cKO mice

Reduced epididymal sperm number and aberrant sperm morphology in *Dnali1* cKO mice suggested impaired spermatogenesis. Therefore, histology of the testes and epididymides in adult control and *Dnali1* mutant mice was examined in 4% paraformaldehyde (PFA) fixed tissues. In control mice, the seminiferous tubule epithelium showed elongated spermatid heads embedded as bundles among the round spermatids, with developing tails extending into the lumen in stages I–III (*Figure 7Aa*). *Dnali1* cKO mice exhibited abnormal elongating spermatid heads with compressed shapes and failure to form elongated bundles in stages I–III (*Figure 7Ab and c*). In stage XII, the control seminiferous tubule epithelium showed step 12 elongating spermatid bundles with large pachytene spermatocytes in meiotic division (*Figure 7Ad*). However, in *Dnali1* cKO mice, stage XII showed abnormal elongating spermatid heads without normal bundle formation (*Figure 7Ae and f*). Additionally, normal spermiation was observed in stage VIII–IX seminiferous tubules in the control mice (*Figure 7Ag*); however, in *Dnali1* cKO mice, evidence of spermiation failure was detected by the phagocytosis of thin heads of step 16 spermatids within the epithelium (*Figure 7Ah and i*). These changes were also demonstrated by periodic acid–Schiff (PAS) staining (*Figure 7—figure supplement 1*) and histological analysis after fixation with Bouin's solution (*Figure 7—figure supplement 2*).

Numerous mature sperm were concentrated in the cauda epididymis of control mice (*Figure 7B*, left). In *Dnali1* cKO mice, the cauda epididymal lumen contained fewer sperm, along with abnormal sperm heads and tails and sloughed round spermatids (*Figure 7B*, right).

## The manchette architecture was not dramatically disrupted in the absence of DNALI1

DNALI1 is present in the manchette. In order to investigate the effect of DNALI1 deficiency in male germ cells on manchette architecture, a pairwise comparison of manchette organization was made on isolated germ cells (*Figure 8*) and testicular sections (*Figure 8—figure supplement 1*) between the control and *Dnali1* cKO mice. In the absence of DNALI1, manchette organization appears to be unaltered.

## Ultrastructural changes of developing germ cells in the seminiferous tubules of the *Dnali1* cKO mice

To investigate the structural basis for the molecular changes observed in the absence of DNALI1, the ultrastructure of testicular seminiferous tubules and cauda epididymal sperm cells was examined. In control mice, the developed sperm nuclei had normal elongated shapes with condensed chromatin (*Figure 9A*), and the flagella contain a normal '9+2' axoneme structure surrounded by accessory structures, including the mitochondrial sheath (*Figure 9B*) and a fibrous sheath (*Figure 9C*). In contrast, in *Dnali1* cKO mice, abnormal shapes of the developed sperm nuclei were frequently observed (*Figure 9D*) and retained cytoplasmic components (not fully resorbed as part of the residual body) were present in the developed sperm (*Figure 9E*, *Figure 9—figure supplement 1A*). Importantly, the two longitudinal columns were usually not associated with microtubule doublets 3 and 8, and the two semicircumferential ribs showed defective or asymmetric formations in the principal piece of the flagellum. Some sperm also had abnormal cell membranes (*Figure 9F*, *Figure 9—figure supplement 1B, C*) and the core '9+2' axoneme was incomplete or disorganized in some sperm (*Figure 9G–I*).

## Inactivation of *Dnali1* in male germ cells did not change protein levels but changed the localization of the downstream proteins

To examine if the loss of DNALI1 in male germ cells affected protein levels of MEIG1, PACRG, and SPAG16L, western blotting was conducted using the testis lysates of control and *Dnali1* cKO mice. There was no significant difference in protein levels of MEIG1, PACRG, and SPAG16L between the control and *Dnali1* cKO mice (*Figure 10A*).

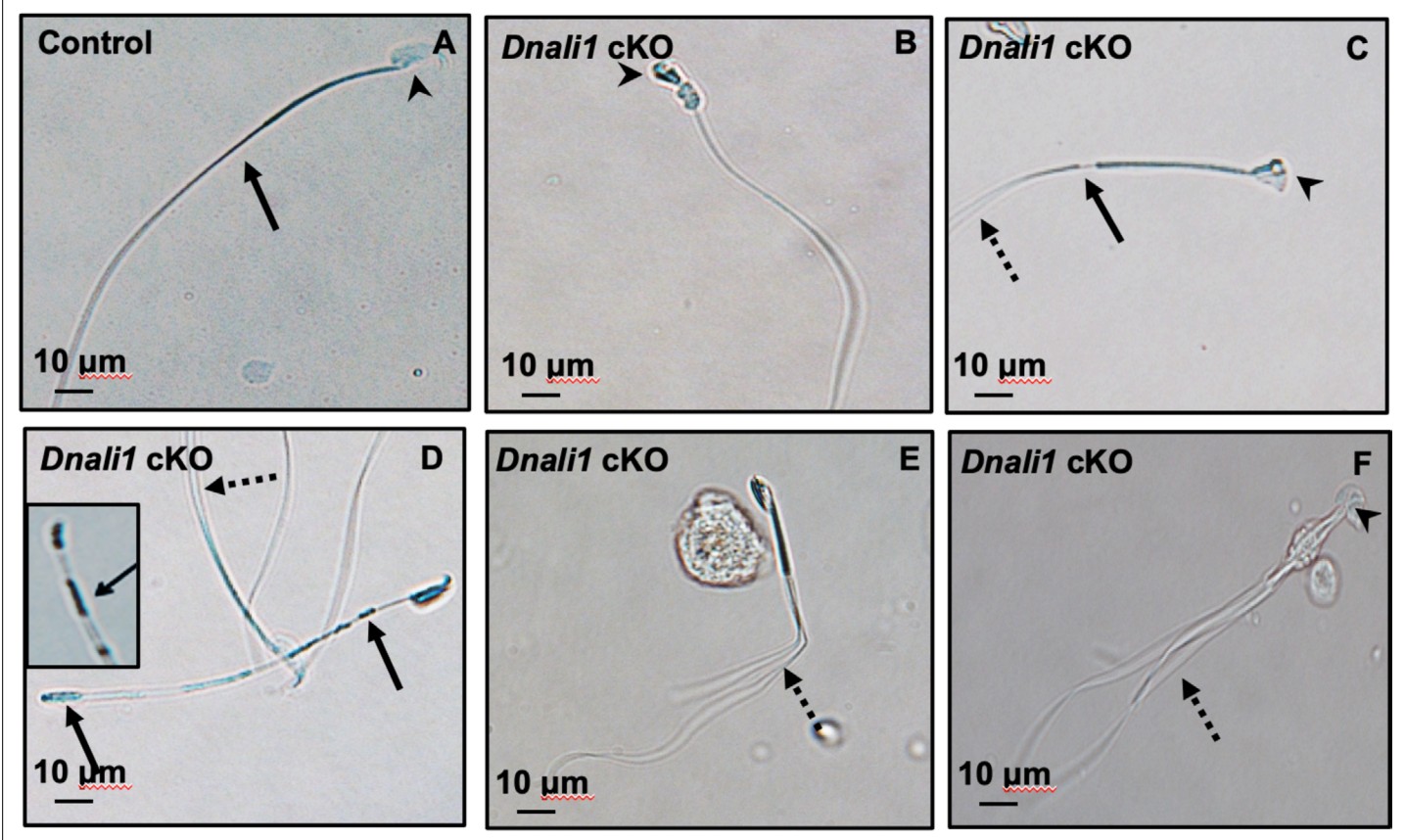

**Figure 6.** Abnormal sperm morphologies in *Dnali1* conditional knockout (cKO) mice. Representative epididymal sperm of control (**A**) and *Dnali1* cKO mice (**B–F**) examined by DIC microscopy. Sperm in the control mice showed normal head (**A**, arrowhead) and flagella (**A**, arrow). Multiple abnormalities were observed in *Dnali1* cKO mice, including distorted heads (**B, C, and F**, arrowheads), uneven thickness tails (**C, D**, arrows). Some sperm showed multiple flagella (**C, D, E, F**, dashed arrows).

The online version of this article includes the following figure supplement(s) for figure 6:

**Figure supplement 1.** Morphological examination of epididymal sperm by light microscopy at high magnification.

Localization of MEIG1 and SPAG16L was further examined. When analyzed by a fluorescence microscope, MEIG1 was present in the cell bodies of spermatocytes and round spermatids of control mice, and protein localization was not changed in the two cell types in *Dnali1* cKO mice (*Figure 10B*). Similarly, SPAG16L was present in the cytoplasm of spermatocytes and round spermatids in both control and *Dnali1* cKO mice (*Figure 10C*). In contrast, while MEIG1 and SPAG16L were present in elongating spermatids of *Dnali1* cKO mice, they appeared not co-localized in the manchette as observed in control mice (*Figure 10D*). To note, in some elongated spermatids, it seems that a weak MEIG1 signal was still present in the manchette when the cells were examined by a confocal microscope (*Figure 10—figure supplement 1*).

## Discussion
### DNALI1 interacts with PACRG

The *Parcg* gene is a reverse strand gene located upstream of the *Parkin* gene, involved in Parkinson's disease (*West et al., 2003*). Genetic disruption of *Pacrg* in mouse phenocopies the infertility phenotype of *Meig1* knockout mice (*Li et al., 2015*; *Li et al., 2021*; *Bennett et al., 1971*; *Lockhart et al., 2004*; *Lorenzetti et al., 2004*; *Taylor et al., 2007*). Our previous studies demonstrated that MEIG1 and PACRG form a complex in the manchette to transport protein cargo and build the sperm flagellum. We further discovered that four amino acids on the same surface of MEIG1 protein are involved in its interaction with PACRG (*Li et al., 2016*; *Li et al., 2021*). However, it was not clear

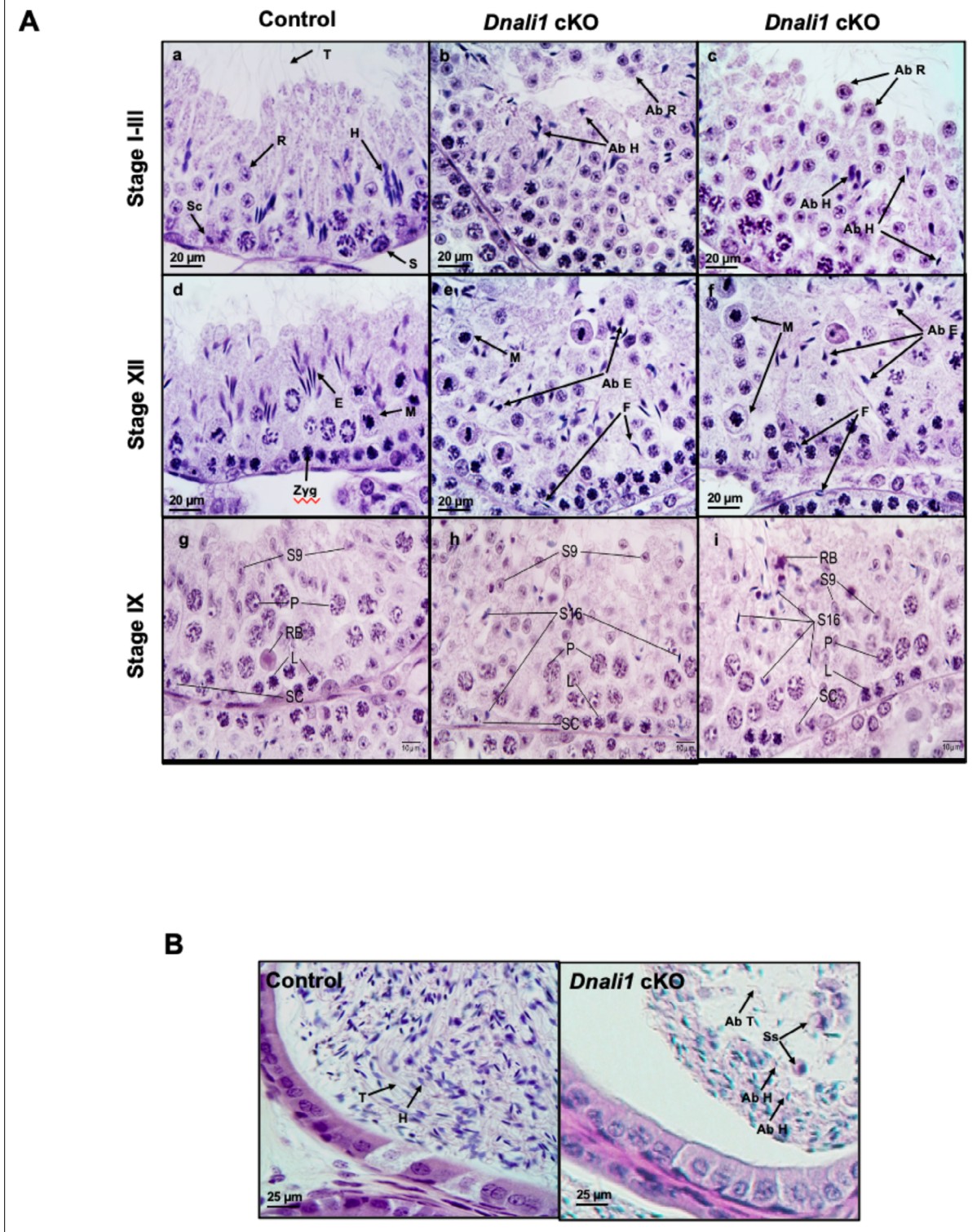

**Figure 7.** Histological analysis for the adult control and *Dnali1* conditional knockout (cKO) mice. (**A**) Histological evaluation of testes from control and *Dnali1* cKO mice, with selected images from stages I-III (**a–c**), stage XII (**d–f**), and stage IX (**g–i**). (**a**) Control testis seminiferous tubule epithelium showing elongated spermatid heads (H) embedded as bundles among the round spermatids (R) and developing tails (T) extending into the lumen. Sc, Sertoli cell; S, spermatogonia. (**b–c**) *Dnali1* cKO with abnormal elongating spermatid heads (Ab H) with compressed shapes and failure to form elongated bundles. Round spermatids are seen sloughing abnormally into the lumen (Ab R). (**d**) Control showing step 12 elongating spermatid bundles (**E**) with large pachytene spermatocytes in meiotic division (**M**). Zygotene spermatocyte, Zyg. (**e–f**) *Dnali1* cKO with abnormal elongating spermatid heads (Ab E) without normal bundle formation. There is evidence of failure of spermiation (**F**), as phagocytosis of thin heads of step 16 spermatids are also present

*Figure 7 continued on next page*

*Figure 7 continued*

in the epithelium. Normal meiotic figures (**M**) are present in stage XII. (**g**) Control seminiferous tubule, with step 9 spermatids (**S9**) lining the lumen and no step 16 elongated spermatids present after spermiation. P, pachytene spermatocytes; L, leptotene spermatocytes; RB, residual body; SC, Sertoli cell. (**h**) *Dnali1* cKO tubule showing step 9 spermatids forming (**S9**) along with step 16 elongated spermatids (**S16**) remaining at various levels within the seminiferous epithelium after failure to spermiate. P, pachytene spermatocytes; L, leptotene spermatocytes; SC, Sertoli cell. (**i**) A second example of a *Dnali1* cKO tubule showing step 9 spermatids (**S9**) but numerous step 16 elongated spermatids (**S16**) that failed to be released into the lumen during spermiation. P, pachytene spermatocytes; L, leptotene spermatocytes; RB, residual body; SC, Sertoli cell. (**B**) Representative histology of epididymis. The control cauda epididymis shows the lumen filled with normal sperm heads (**H**) and tails (**T**). In *Dnali1* cKO male, the cauda epididymal lumen contains numerous sperm with abnormal heads (Ab H) and tails (Ab T) and sloughed round spermatids (Ss).

The online version of this article includes the following figure supplement(s) for figure 7:

**Figure supplement 1.** Testicular histology of adult control and *Dnali1* conditional knockout (cKO) mice examined by periodic acid–Schiff (PAS) staining.

**Figure supplement 2.** Testicular histology of adult control and *Dnali1* conditional knockout (cKO) mice examined by hematoxylin and eosin (H&E) staining.

how the MEIG1/PACRG complex associates with the manchette microtubules. Given that MEIG1 is a downstream binding partner of PACRG, and the fact that PACRG does not associate with microtubules directly, we hypothesized that there must be other upstream players involved in the manchette localization of the MEIG1/PACRG complex. Therefore, we performed a yeast two-hybrid screen using a PACRG component as bait. Besides MEIG1, DNALI1 was identified to be a major binding partner. The interaction between PACRG and DNALI1 was further confirmed by several experiments described in this study, including co-expression and purification from bacteria and co-immunoprecipitation using mouse testis protein extracts. Interestingly, testicular PACRG protein level was not changed in the *Dnali1*^*Stra8KO*^ mice, which would suggest that PACRG also associates with other proteins, including MEIG1, and these proteins also stabilize PACRG even if DNALI1 is absent.

The functional assays reported here also support the observed interaction between PACRG and DNALI1. Previously, we discovered that mouse PACRG was not stable but could be stabilized by MEIG1 in both bacteria and mammalian cells (*Li et al., 2016*). The studies here showed that DNALI1 also stabilized PACRG, suggesting a dual association of PACRG with MEIG1 and DNALI1. Similar to

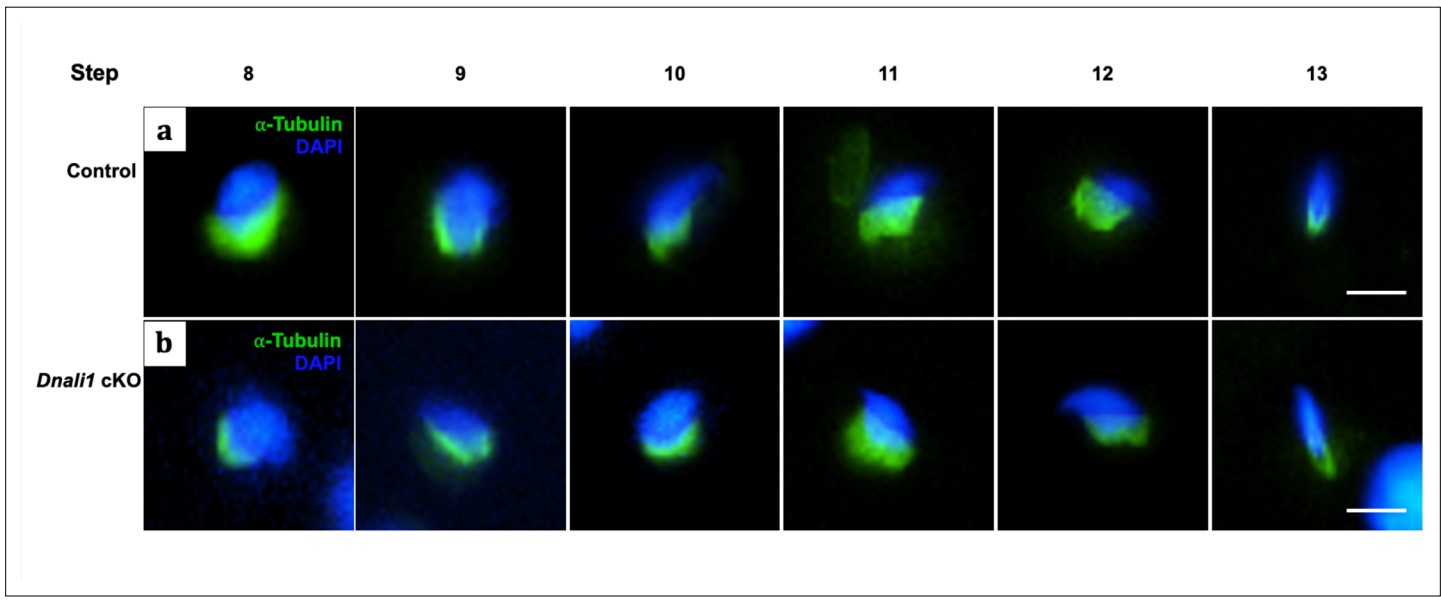

**Figure 8.** The manchette architecture is similar between the control and *Dnali1* conditional knockout (cKO) mice in isolated germ cells. The testicular cells were isolated from the control and *Dnali1* cKO mice, and immunofluorescence staining was conducted using an anti-α-tubulin (green) antibody. The nuclei were stained with DAPI (blue). Notice that the distribution of α-tubulin in elongating spermatids was similar between control (**a**) and *Dnali1* cKO (**b**) mice. Scale bar: 5 μm. Images were taken under a Nikon DS-Fi2 Eclipse 90i Motorized Upright Fluorescence Microscope (The C.S. Mott Center for Human Growth and Development, Department of Obstetrics & Gynecology, Wayne State University).

The online version of this article includes the following figure supplement(s) for figure 8:

**Figure supplement 1.** Examination of manchette structure in the testis seminiferous tubules of the control and *Dnali1* conditional knockout (cKO) mice.

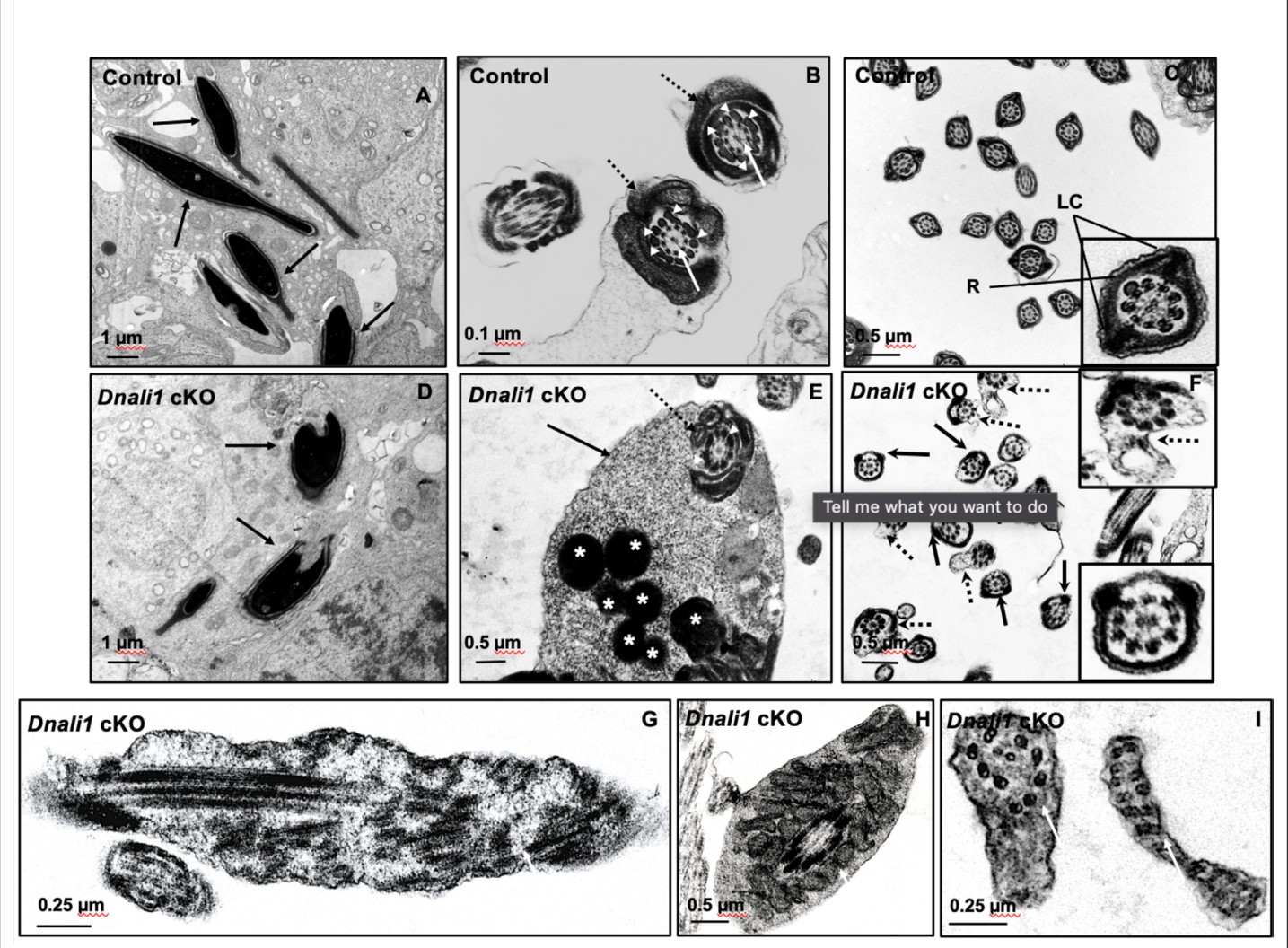

**Figure 9.** Ultrastructural changes of testicular sperm of control and *Dnali1* conditional knockout (cKO) mice. The ultrastructure of testicular sperm from the control (**A, B, and C**) and *Dnali1* cKO (**D–I**) mice were analyzed by transmission electron microscopy (TEM). (A) Control testis seminiferous tubule epithelium showing nuclei with normally condensed chromatin (arrows). (**B**) Control mouse showed the normal midpiece of flagella with normal '9+2' axoneme structure in the center (white arrows) surrounded by mitochondrial sheath (black dotted arrows) and ODF (white arrow heads). (**C**) Normal principal piece of flagella from a control mouse, which is characterized by the presence of a complete fibrous sheath surrounding the axoneme. The fibrous sheath consists of two longitudinal columns (LC) connected by semicircumferential ribs (**R**). The two longitudinal columns are associated with microtubule doublets 3 and 8, and the two semicircumferential ribs are symmetrical. (**D**) Abnormally condensed chromatin in *Dnali1* cKO mouse (black arrows). (**E**) A representative image of the midpiece in a flagellum from a *Dnali1* cKO mouse. The ODF (white arrow heads) and mitochondrial sheath (black dotted arrow) were present, but cytoplasm residue (black arrow) remained with a number of the lysosomes inside (white stars). (**F**) The flagella show disorganized fibrous sheath structure in the *Dnali1* cKO mouse. Noticed that the two longitudinal columns are not associated with microtubule doublets 3 and 8 in some flagella, and the two semicircumferential ribs showing defective or asymmetric organization (black arrows and the lower, right insert); some flagella also have disrupted membranes (dashed arrows and upper, right insert). (**G–I**) The flagella show disrupted axonemes in *Dnali1* cKO mice (white arrows).

The online version of this article includes the following figure supplement(s) for figure 9:

**Figure supplement 1.** Additional testicular sperm transmission electron microscopy (TEM) images of the *Dnali1* conditional knockout (cKO) mice.

the MEIG1 protein, co-expression of DNALI1 also prevented degradation of PACRG in both bacteria and mammalian cells, which is consistent with the reported regulation of PACRG by the ubiquitin-proteasomal system (UPS) (*Taylor et al., 2012*). It is possible that the binding site on the PACRG recognized by the UPS is protected by DNALI1, providing PACRG protein more stability.

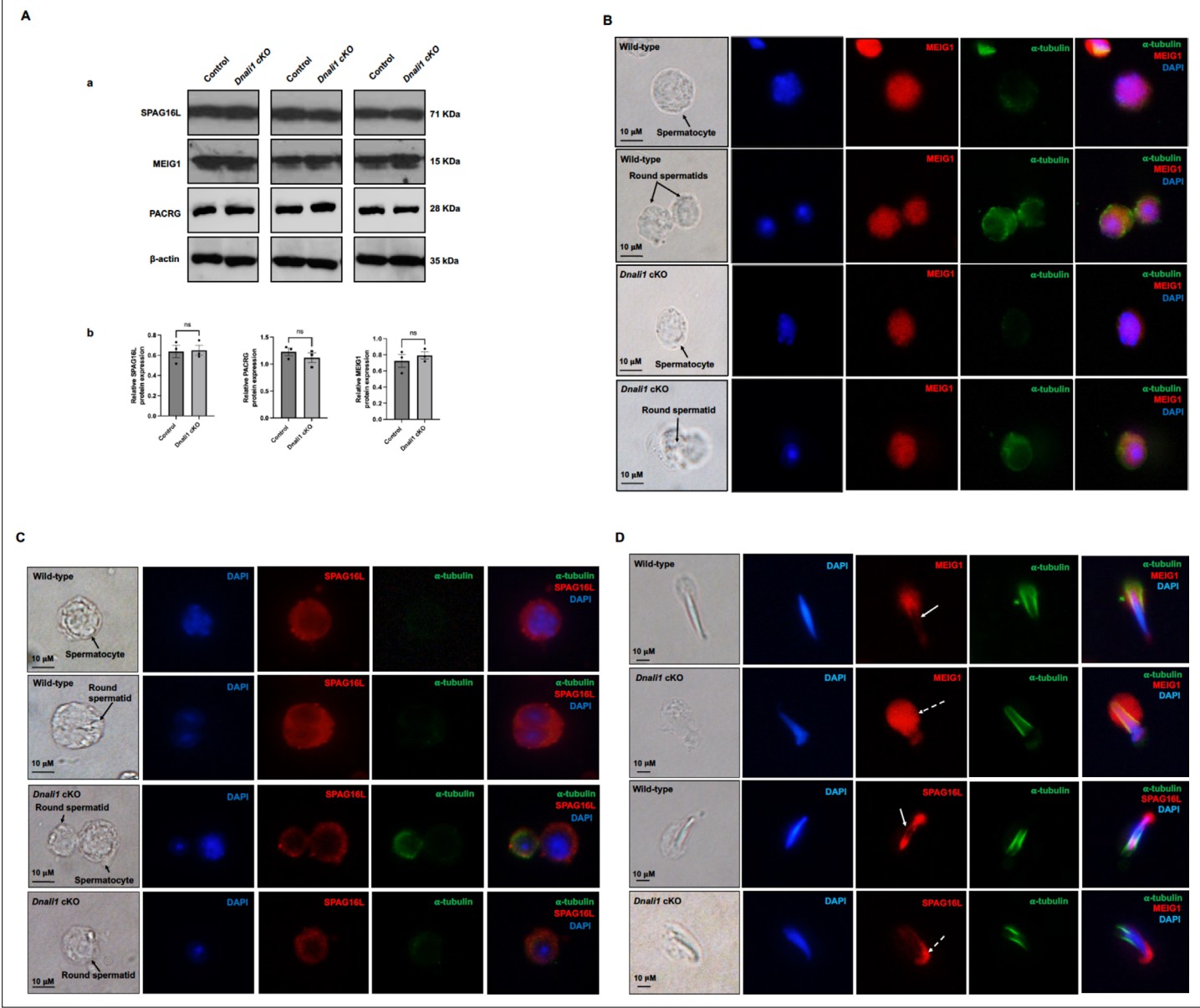

**Figure 10.** Expression levels and localization of DNALI1 downstream proteins in the *Dnali1* conditional knockout (cKO) mice. (**A**) Analysis of testicular expression of MEIG1, PACRG, and SPAG16L in control and *Dnali1* cKO mice by western blot. Compared with control mice, there was no significant change in the expression level of these three proteins in *Dnali1* cKO mice. (**a**) Representative western blot results. (**b**) Statistical analysis of the protein levels normalized by β-actin. n=4. (**B**) Localization of MEIG1 in spermatocytes and round spermatids of the control and *Dnali1* cKO mice by immunofluorescence staining. There is no difference between the control and *Dnali1* cKO mice. MEIG1 is present in cell bodies in both genotypes. (**C**) Localization of SPAG16L in spermatocytes and round spermatids of the control and *Dnali1* cKO mice by immunofluorescence staining. There is no difference between the control and the *Dnali1* cKO. SPAG16L is present in cell bodies in both genotypes. The white dashed arrow points to an elongating spermatid, and the SPAG16L is present in the manchette. (**D**) Localization of MEIG1 and SPAG16L in elongating spermatids of the control and *Dnali1* cKO mice. Both MEIG1 and SPAG16L are present in the manchette in the control mice (white arrows); however, they are no longer present in the manchette in *Dnali1* cKO mice (dashed white arrows). Images were taken under a Nikon DS-Fi2 Eclipse 90i Motorized Upright Fluorescence Microscope (The C.S. Mott Center for Human Growth and Development, Department of Obstetrics & Gynecology, Wayne State University).

The online version of this article includes the following source data and figure supplement(s) for figure 10:

**Source data 1.** Expression levels of DNALI1 downstream proteins in the *Dnali1* conditional knockout (cKO) mice.

**Figure supplement 1.** Examination of MEIG1 localization in elongating spermatids in the control and *Dnali1* conditional knockout (cKO) mice.

This association between PACRG and DNALI1 was also supported by in vivo studies. During the first wave of spermatogenesis, expression of the DNALI1 protein was dramatically increased during the spermiogenesis phase, which was similar to the PACRG protein (*Rashid et al., 2006*; *Li et al., 2016*) and specific to the period of spermatid elongation and formation of flagella. However, prior to spermiogenesis, in late meiotic germ cells, the DNALI1 signal was weak (*Rashid et al., 2006*). In elongating spermatids, DNALI1 was localized in the manchette, coinciding with the localization of PACRG, which strongly suggested that DNALI1 would have a function related to that of PACRG. Testing this hypothesis through the use of conditional *Dnali1* knockout mice revealed a co-dependence between the two proteins for structural integrity of spermatid elongation, the absence of which leads to abnormal sperm morphology, immotility, and complete male infertility.

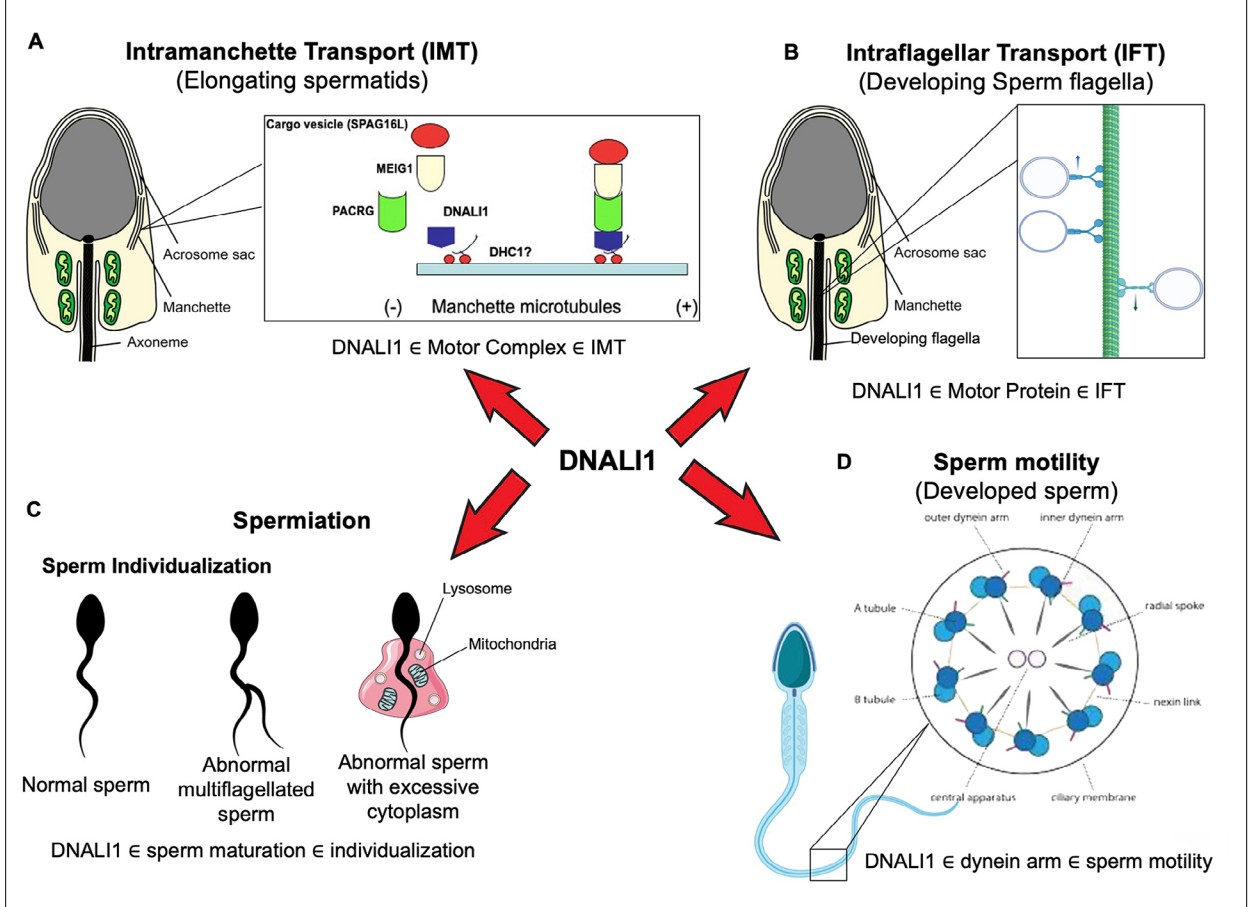

**Figure 11.** Working model of DNALI1 in sperm cell differentiation and function. (**A**) DNALI1 forms a complex with MEIG1/PACRG, with DNALI1 being an upstream protein that recruits downstream PACRG and MEIG1 to the manchette. DNALI1 associates with the manchette microtubules through other molecular motor protein(s), including dynein heavy chain 1. MEIG1/PACRG/DNALI1/motor complex transports cargos, including SPAG16L, along the manchette to build sperm flagella. (**B**) DNALI1 might also function as a motor protein involved in transporting intraflagellar transport (IFT) particles. (**C**) DNALI1 may facilitate in the appropriate maturation and individualization of sperm cells. (**D**) DNALI1 is present in the dynein arm and functions in sperm motility.

The online version of this article includes the following source data and figure supplement(s) for figure 11:

**Figure supplement 1.** Examination of dynein heavy chain 1 protein (DYNC1H1) in male germ cells by immunofluorescence staining.

**Figure supplement 2.** Basal body/centriole formation was not affected in the absence of DNALI1 in male germ cells.

**Figure supplement 3.** DNALI1 is present in the developing mouse sperm flagella.

**Figure supplement 4.** Inactivation of *Dnali1* in male germ cells resulted in abnormal autophagy pathway.

**Figure supplement 4—source data 1.** Western blot analysis of selective core autophagy components, ATG16L and ATG5.

## DNALI1 is involved in IMT

It is not surprising to see dramatically reduced sperm counts associated with impaired spermiogenesis, because DNALI1 forms a complex with MEIG1/PACRG. Disruption of either *Meig1* or *Pacrg* in the male germ cells resulted in a similar phenotype of impaired spermiogenesis and male infertility (*Zhang et al., 2009*; *Li et al., 2015*; *West et al., 2003*; *Bennett et al., 1971*; *Lockhart et al., 2004*; *Lorenzetti et al., 2004*; *Taylor et al., 2007*). The manchette is believed to play an important role in transporting cargo proteins for the formation of sperm flagella, and the transport function requires motor proteins (*Kierszenbaum, 2002*; *Lehti and Sironen, 2016*). Several motor proteins have been localized in the manchette (*Kierszenbaum et al., 2003*; *Yang et al., 2006*; *Hayasaka et al., 2008*; *Saade et al., 2007*), and some have been shown to be essential for flagella formation (*Lehti et al., 2013*; *Christman et al., 2021*). It is highly possible that DNALI1 is the major driving force in transporting the MEIG1/PACRG complex together with the sperm flagellar proteins, including SPAG16L, along the manchette microtubules in order to assemble a functional sperm tail. It appears that DNALI1 is an upstream protein of MEIG1/PACRG complex, because DNALI1 is still in the manchette when PACRG is absent (*Figure 11A*).

In the absence of DNALI1, the manchette localization of MEIG1 and the cargo protein SPAG16L was greatly affected. However, a trace amount of MEIG1 was observed in the manchette when *Dnali1* was inactivated in male germ cells. One possible explanation is that *Dnali1* was not 100% disrupted due to the efficiency using the cKO system. However, it is also possible that some of the MEIG1 does not associate with the DNALI1 complex, which is supported by the fact that slightly different phenotypes were observed in the two knockout models. For example, the manchette architecture was abnormal in the *Meig1* knockout mice (*Zhang et al., 2009*), but not in the *Dnali1* cKO mice. In the manchette, there is also an F-actin-based myosin motor transport pathway (*Kierszenbaum et al., 2003*), which may also be associated with some MEIG1. It remained to be known if PACRG localization is changed in the *Dnali1* cKO mice. Our supply was depleted of the two anti-PACRG antibodies that worked in the immunofluorescence experiments (the polyclonal antibody from Baylor College of Medicine) (*Zhang et al., 2009*; *Prevo et al., 2017*) and the monoclonal antibody generated in our laboratory (*Li et al., 2015*). Two other commercial antibodies (Cat No: ARP52953_P050, Aviva Systems Biology, San Diego, CA, USA; Cat No: bs-9289R, Bioss, Woburn, MA, USA) unfortunately gave no specific signals. Therefore, further investigation should be performed once appropriate antibodies are available. We expect that PACRG could be mislocalized in the absence of DNALI1.

As a light chain protein, it is unlikely that DNALI1 binds directly to the manchette microtubules, based on its localization in the transfected mammalian cells. DNALI1 has been reported to be a binding partner of dynein heavy chain 1 protein (DYNC1H1) (*Rashid et al., 2006*) and our current study revealed evidence that the dynein heavy chain 1 protein is also present in the manchette (*Figure 11—figure supplement 1*). Thus, it is more likely that DNALI1 associates with the manchette microtubules through dynein heavy chain 1. Another function of the manchette is to help shape the head of developing spermatids (*Kierszenbaum and Tres, 2004*; *Manfrevola et al., 2021*). Like the *Meig1* and *Pacrg* mutant mice, the *Dnali1* cKO mice also developed abnormal sperm heads, which further supports the claim that DNALI1 is involved in manchette function, but not necessarily in manchette construction, which appeared to have normal morphology.

Formation of the flagellum involves both the IMT and the IFT, and the abnormal sperm phenotype observed in *Dnali1* cKO mice appears to be due to the disruption of both pathways. The IMT mediates the transport of cargo proteins to the basal bodies, the template and the start point from which the sperm axoneme is formed (*Kierszenbaum, 2002*). In *Dnali1* cKO mice, although some sperm were formed, none had normal morphology. Besides abnormal heads, the flagella showed multiple defects, including short tails and vesicles and gaps along the sperm tail. The partial formation of the tails indicates that the IMT may not always be disrupted, possibly due to an incomplete excision by the Cre recombinase, allowing for limited DNALI1 protein synthesis; or the cargo proteins are transported through the myosin-based transport system. It is unlikely that abnormal tail formation was caused by the defect of basal bodies. In the absence of DNALI1 in male germ cells, the basal body formation appeared to be normal (*Figure 11—figure supplement 2*).

## Potential role of DNALI1 as a motor protein involved in transporting IFT particles

The IFT is one of the more important mechanisms involved in the formation of cilia/flagella (*Rosenbaum and Witman, 2002*). The IFT complex is composed of core IFT components, BBSomes, the motor proteins and cargo proteins (*Prevo et al., 2017*). The morphological abnormalities observed in *Dnali1* cKO sperm, including the trapped particles inside sperm tails, led us to hypothesize that DNALI1 may also be involved in the IFT process, which is supported by the presence of DNALI1 in the developing sperm flagella (*Figure 11—figure supplement 3*). Defects in IFT result in failure of ciliogenesis, including formation of the sperm flagellum (*Pazour and Rosenbaum, 2002*; *Zhang et al., 2016*; *Liu et al., 2017*; *Zhang et al., 2017*; *Shi et al., 2019*; *Qu et al., 2020*; *Zhang et al., 2018*; *Zhang et al., 2020*). Motor proteins have been shown to be essential for a functional IFT (*Scholey, 2013*; *Scholey, 2008*). Therefore, given the ultrastructural abnormalities in the sperm flagella, loss of DNALI1 might also result in IFT dysfunction (*Figure 11B*), and be a major contributing factor to sperm immotility. Although we cannot exclude the possibility that other transport systems mediate the transport of cargo proteins in the manchette, data presented here show that the MEIG1/PACRG/DNALI1 complex does transport some cargo proteins that are essential for the formation of a normal sperm tail.

Transmission electron microscopy (TEM) study revealed a disorganized fibrous sheath, sperm tail structure in mice. The fibrous sheath is a cytoskeletal structure of the principal piece in the flagellum, which consists of two longitudinal columns connected by semicircular ribs. The longitudinal columns are attached to outer dense fibers 3 and 8 in the anterior part of the principal piece and replace those fibers in the middle and posterior part of the principal piece and become associated with microtubule doublets 3 and 8 (*Fawcett, 1975*). It is assumed that such an elaborate substructural organization of the fibrous sheath into longitudinal columns and semicircular ribs must have a function, but testable hypotheses surrounding these characteristics are lacking. In *Dnali1* cKO mice, the two longitudinal columns were not associated with microtubule doublets 3 and 8, and the two semi-circumferential ribs showed defective or asymmetrical alignment in the principal piece. Dysfunction in IMT and IFT may contribute to these structural defects.

## DNALI1 may facilitate in the appropriate maturation and spermiation/individualization of sperm cells

Another interesting phenotype observed was an occasional failure of sperm spermiation/individualization, or disengagement (*O'Donnell and Stanton, 2018*; *Hess and Vogl, 2015*), at the late spermatid step in *Dnali1* cKO mice. This abnormality was not observed in the *Meig1* and *Pacrg* mutant mice (*Zhang et al., 2009*; *Li et al., 2015*; *West et al., 2003*; *Bennett et al., 1971*; *Lockhart et al., 2004*; *Lorenzetti et al., 2004*; *Taylor et al., 2007*). The spermiation/individualization failure may be attributed to the dysregulation of autophagy in the absence of DNALI1, as suggested by the upregulation of ATG16L and ATG5, two markers of the autophagic pathway, in *Dnali1* cKO mice (*Figure 11—figure supplement 4*). Spermiation/individualization is one of the final steps of spermiogenesis and the defects are a common cause of human male infertility (*Cooper, 2005*). In *Drosophila*, this occurs when the membrane cytoskeleton individualized complex (IC) is assembled around the nucleus of each bundle of 64 haploid elongated spermatid nuclei. Each IC is composed of 64 F-actin-based investment cones, which move the flagella downward as a coordinated ensemble. Each spermatid is individualized of a single cone. As IC progresses, the cytoplasm is removed from the flagella, and the membrane surrounding each spermatid is reshaped to form the individualized spermatozoa. In mammals, the process is thought to be driven by tubulobulbar complexes (TBC) and involves the removal of ectoplasmic specializations, and actin filament networks are also the major components of TBCs (*Hess and Vogl, 2015*; *Cooper, 2005*; *O'Donnell et al., 2011*; *Tokuyasu et al., 1972*; *Adams et al., 2018*). It has been reported using genetic studies that at least 70 genes are involved in the process of spermiation/individualization, with a unique cooperation between Sertoli cells and the maturing elongated spermatids (*Hess and Vogl, 2015*; *Steinhauer, 2015*). Several molecular pathways contribute to this process, including actin and microtubule dynamics, the ubiquitin-proteasome pathway components, apoptotic elimination of cytoplasmic contents, plasma membrane reorganization, and the formation of a disengagement complex (*O'Donnell and Stanton, 2018*; *Hess and Vogl, 2015*; *Steinhauer, 2015*; *Zhong and Belote, 2007*; *Chen et al., 2021*). The microtubule cytoskeleton

is important for spermiation/individualization (*Steinhauer, 2015*). Complexes of microtubules and dynein and kinesin motors form the core of cilia and flagella (*Sweeney and Holzbaur, 2018*). Mutations in the components of the dynein-dynein complex, including the cytoplasmic dynein intermediate chain, the two *Drosophila* dynein light chains DDLC1 and DLC90F, disrupt the synchronous movement of actin cones, but they also disrupt the nucleus shaping and positioning (*Ghosh-Roy et al., 2005*; *Ghosh-Roy et al., 2004*; *Li et al., 2004*). These consistent phenotypes suggest that DNALI1 may have a similar effect on the synchronous movement of actin cones in sperm spermiation/individualization and that DNALI1 has other functions in male germ cell development that differ from the MEIG1/PACRG complex (*Figure 11C*). Consistent with this, our recent yeast two-hybrid screen using human DNALI1 as the bait, γ-actin identified to be a potential binding partner, further supports that DNALI1 is involved in spermiation/individualization.

## DNALI1 is present in the dynein arm and functions in sperm motility

Another potential function of DNALI1 is its reported activity as an IDA component of the cilium and flagellum axoneme (*Kamiya and Okamoto, 1985*). This would not be surprising in sperm, as it is present along the entire length of the sperm flagellar axoneme (*Rashid et al., 2006*). Dynein proteins are present in the outer dynein arms (ODAs) and IDAs of the axonemal complex, and both arms are essential for the beating of cilia and flagella (*Kamiya and Okamoto, 1985*; *Mitchell and Rosenbaum, 1985*). IDAs have seven major subspecies and four minor subspecies (*Scholey, 2008*), but little is known about the functional differences between these subspecies (*Bui et al., 2012*; *Yagi et al., 2005*; *Kamiya and Yagi, 2014*). Therefore, if DNALI1 does serve this function in the sperm axoneme, its inactivation in male germ cells would likely disrupt the function of the dynein arms and contribute to the formation of immotile sperm (*Figure 11D*).

In summary, DNALI1 potentially has multiple roles in sperm formation and function. We demonstrated a main function in IMT but additional roles of DNALI1 are possible in IFT and sperm individualization and disengagement, all of which being biological processes that occur during spermiogenesis. Failure of these processes causes impaired sperm formation and function and finally results in male infertility. While we show that an IMT function is clearly associated with the MEIG1/PACRG complex, the potential function of DNALI1 in IFT, sperm individualization, and spermiation will need further investigation.

## Materials and methods
### Cell lines
Cells used in this study were a gift from Dr. Jerome Strauss. Verification of cell line identity was performed through observation of cell morphology and characteristics. CHO cells had adherent, epithelial cell-like appearance. COS-1 cells had adherent, fibroblast-like morphology. HEK293 cells used in the study exhibited adherent, epithelial morphology.

### Yeast two-hybrid experiments
Full-length mouse *Pacrg* coding sequence was amplified using the following primers: forward: 5'-GAATTCATGCCGAAGAGGACTAAACTG-3'; reverse: 5'- GGATCCGTCAGTTCAGCAAGCACGACTC-3'. After TA cloning and sequencing, the correct cDNA was subcloned into the EcoR1/BamH1 sites of pGBKT7, which was used to screen a Mate & Plate Library-Universal Mouse (Normalized) (Clontech, Mountainview, CA, USA; Cat No: 630482) using the stringent protocol according to the manufacturer's instructions. For direct yeast two-hybrid assay, the coding sequence of the mouse *Dnali1* cDNA was amplified by RT-PCR using the following primers: forward: 5'- GAATTCATGATACCCCCAGCAGACTCTCTG-3' and reverse: 5'- GGATCCGATCACTTCTTCGGTGCGATAATGCC-3'. The correct cDNA was cloned into EcoRI/BamH1 sites of pGAD-T7 vector. The yeast was transformed with the indicated plasmids using the Matchmaker Yeast Transformation System 2 (Clontech, Cat No: 630439). Two plasmids containing simian virus 40 large T antigen (*LgT*) in pGADT7 and Trp53 in pGBKT7 were co-transformed into AH109 as a positive control. The AH109 transformants were streaked out in complete drop-out medium (SCM) lacking tryptophan, leucine, and histidine to test for histidine prototrophy.

### Localization assay
To generate mouse PACRG/pEGFP-N$_2$ plasmid, *Pacrg* cDNA was amplified using the primer set: forward: 5'-GAATTCATGCCGAAGAGGACTAAACTG-3; reverse: 5'- GGATCCGGTTCAGCAAGC

ACGACTC-3', and the correct *Pacrg* cDNA was ligated into the pEGFP-N$_2$ vector. To generate mouse DNALI1/Flag construct, *Dnali1* cDNA was amplified using the primer set: forward: 5'-<u>GAATTC</u>AATG ATACCCCCAGCAGACTCTCTG-3'; reverse: 5'-<u>CTCGAG</u>TCACTTCTTCGGTGCGATAATGCC-3', and the correct *Dnali1* cDNA was ligated into the pCS3+ FLT vector. The PACRG/pEGFP-N$_2$ and DNALI1/ Flag were transfected individually or together into CHO cells which show clear morphology by using Lipofectamine 2000 transfection reagent (Invitrogen, Waltham, MA, USA). The CHO cells were cultured with DMEM (with 10% fetal bovine serum) at 37°C. After 48 hr post transfection, the CHO cells were processed for immunofluorescence with an anti-Flag antibody. Images were taken with a confocal laser scanning microscopy (Zeiss LSM 700, Virginia Commonwealth University).

## Co-immunoprecipitation assay

Mouse *Pacrg* cDNA was amplified using the following primers: forward; 5'-<u>GAATTC</u>ACCAGACA AGATGCCGAAGAGG-3'; reverse: 5'- <u>TCTAGA</u>GGTCAGTTCAGCAAGCACGACTC-3', and the correct *Pacrg* cDNA was ligated into the pCS2+ MT vector to create the PACRG/Myc plasmid. The PACRG/Myc and DNALI1/Flag plasmids were co-transfected into COS-1 cells which express high level of the exogenous genes by using Lipofectamine 2000 transfection reagent (Invitrogen). After 48 hr post transfection, the cells were processed for co-immunoprecipitation assay. For co-immuno-precipitation assays, the cells were lysed with IP buffer (Beyotime, Jiangsu, China; Cat No. P0013) for 5 min and centrifuged at 10,000 × *g* for 3–5 min. For in vivo co-immunoprecipitation assay, testes from wild-type adult mice were homogenized using immunoprecipitation buffer (150 mM NaCl/50 mM Tris–HCl, pH 8.0/5 mM EDTA/1% Triton X-100/1 mM PMSF/proteinase inhibitor mixture), and the lysates were passed through a 20-gauge needle, followed by centrifugation at 11,600 × *g* for 5 min. The supernatant was pre-cleaned with protein A beads at 4°C for 30 min, and the pre-cleared lysate was then incubated with an anti-MYC antibody (cells) and an anti-PACRG antibody (testis) at 4°C for 2 hr. The mixture was then incubated with protein A beads at 4°C overnight. The beads were washed with IP buffer three times and then re-suspended in 2× Laemmli sample buffer and heated at 95°C for 5 min. The samples were centrifuged at 3000 × *g* for 30 s, and the supernatant was then subjected to western blot analysis with anti-MYC and Flag antibodies (cells) and an anti-DNALI1 antibody (testis).

## Co-expression of mouse PACRG and DNALI1 in bacteria

Full-length mouse *Pacrg* cDNA (amplified by a forward primer: 5'-<u>GAATTC</u>GGTGCCGCGCGGCAGC ATGCCGAAGAGGACTAAACTG-3' and a reverse primer: 5'-<u>GTCGAC</u>TCAGTTCAGCAAGCACGACT C-3') was cloned into the upstream multiple clone site of the dual expression vector pCDFDuet-1 to create the PACRG/pCDFDuet-1 plasmid, and the translated protein was tagged with hexahistidine. The full-length mouse *Dnali1* cDNA (amplified by a forward primer: 5'-<u>GATATC</u>GATGATACCCCCAGCA GACTCTC-3' and a reverse primer: 5'-<u>CTCGAG</u>TCACTTCTTCGGTGCGATAATG-3') was inserted into the downstream multiple cloning site to create the PACRG/DNALI1/pCDFDuet-1 plasmid. The dual expression plasmid was transformed into the Rosetta II (DE3) (Invitrogen) *Escherichia coli* strain, grown in Luria Bertani medium, and induced with 1 mM isopropyl-β-d-thiogalactopyranoside at an A600 ~0.8 for 2 hr. The bacterial pellets from 1 L of growth media were re-suspended in 30 mL of B-PER reagent (Thermo Scientific, Waltham, MA, USA), and co-expressed proteins were purified from the lysis super-natant by nickel affinity chromatography. The purified protein was analyzed by western blotting.

## Luciferase complementation assay

HEK293 cells were grown in 12-well plates and transfected in triplicate with N- and C-luciferase fragments (supplied by Dr. James G. Granneman, Wayne State University) fused to mouse PACRG and DNALI1 along with various controls as specified in the figure legends. The following primers were used to create the constructs: *Pacrg*/C-Luc forward: 5'-<u>AAGCTT</u>CGATGGTGAAGCTAGCTGCC AAATG-3', *Pacrg*/C-Luc reverse: 5'-<u>ACCGGT</u>GGCACCAGGGTATGGAATATGTCCAC-3', N-Luc/*D-nali1* forward: 5'-<u>AAGCTT</u>ATGGCAGAGTTGGGCCTAAATGAG-3', N-Luc/*Dnali1* reverse: 5'-<u>ACCGGT</u> GGATCTTCAGATTCATATTTTGCCAG-3'. After transfection, cells were cultured for 24 hr. Luciferase activities were measured as described previously (*Huang et al., 2021*) and readings were recorded using Veritas microplate luminometer. Experiments were performed three times independently, and the results are presented with standard errors.

## Western blotting analysis

Tissue samples were collected from 3- to 4-month-old mice, and tissue extracts were obtained after lysis with buffer containing 50 mM Tris–HCl pH 8.0, 170 mM NaCl, 1% NP40, 5 mM EDTA, 1 mM DTT, and protease inhibitors (Complete Mini; Roche Diagnostics GmbH, Basel, Switzerland). Protein concentration for lysates was determined using BCA reagent (Sigma-Aldrich, St. Louis, MO, USA), and equal amounts of protein (50 μg/lane) were heated to 95°C for 10 min in sample buffer, loaded onto 10% sodium dodecyl sulfate-polyacrylamide gels, separated with electrophoresis, and transferred to polyvinylidene difluoride membranes (Millipore, Bedford, MA, USA). Membranes were blocked (Tris-buffered saline solution containing 5% nonfat dry milk and 0.05% Tween 20 [TBST]) and then incubated with the indicated primary antibodies at 4°C overnight. After being washed in TBST, the blots were incubated with immunoglobulin conjugated to horseradish peroxidase for 1 hr at room temperature. After washing, the target proteins were detected with Super Signal chemiluminescent substrate (Pierce, Thermo Scientific). The following primary antibodies were used: anti-His (1: 2000, Cat No: 70796-4, Novagen, Madison, WI, USA); anti-DNALI1 (1: 10,000 for Proteintech 17601-1-AP, Rosemont, IL, USA; and 1:2000 for Abcam, Cambridge, UK, Cat No: ab87075; this antibody has been discontinued); anti-GFP (1: 1000, Cat No: 11814460001, Roche); anti-MYC (1: 2000, Cat No: C-19 Sc788, Santa Cruz, Dallas, TX, USA); β-actin (1: 2000, Cat No: 4967S, Cell Signaling Technology, Danvers, MA, USA); anti-ATG16L (1:1000; Cat No: AP1817b; ABGENT, San Diego, CA, USA) and anti-ATG5 (1:1000, Cat No: 12,994S, Cell Signaling Technology, Danvers, MA, USA). Secondary antibodies include anti-Rabbit IgG (1: 2000, Cat No: 711166152, Jackson ImmunoResearch, West Grove, PA, USA) and anti-Mouse IgG (1: 2000, Cat No: DI-2488, Vector Laboratories, Burlingame, CA, USA). Each western blot was performed using three independent biological replicates.

## Real-time PCR

COS-1 cells were transfected with plasmids to express PACRG-GFP alone or co-express PACRG-GFP and DNALI1-MYC. Forty-eight hr after transfection, the total RNAs were extracted and qPCR was conducted to examine *Pacrg* mRNA expression using the following primers: m*Pacrg*F: 5'-GCCCT ACTACCGTCAGATCC-3', and m*Pacrg*R: 5'-AGCACGACTCATAGGTAGGC-3'. The mRNA levels were normalized with *18s* rRNA (forward: 5'-GGGAGCCTGAGAAACGGC-3' and *18s* rRNA reverse: 5'-G GGTCGGGAGTGGGTAATTT-3').

## Preparation of testicular cells and immunofluorescence analysis

Testes were separated from 3- to 4-month-old mice and gently cut into pieces, followed by incubation in a 15 mL centrifuge tube with 5 mL DMEM containing 0.5 mg/mL collagenase IV and 1.0 μg/mL DNase I (Sigma-Aldrich) for 30 min at 32°C and shaken gently. Then the testes were washed once with phosphate-buffered saline (PBS) after centrifugation at 1000 rpm for 5 min under 4°C, and the supernatant was discarded. Afterward, the cell pellet was fixed with 5 mL PFA containing 0.1 M sucrose and shaken gently for 15 min at room temperature. After washing three times with PBS, the cell pellet was re-suspended with 2 mL PBS and loaded on positively charged slides. The slides were stored in a wet box at room temperature after air drying. Then, the spermatogenic cells were permeabilized with 0.1% Triton X-100 (Sigma-Aldrich) for 5 min at 37°C. Finally, the samples were incubated overnight with primary antibodies (anti-DNALI1: 1: 100, Abcam, Cat No: ab87075; anti-PACRG: 1: 200, generated by our laboratory; anti-α-tubulin, 1:200, Sigma, Cat No: T9026-2mL; anti-MEIG1: 1:400, generated by our laboratory; anti-SPAG16L, 1: 200, generated by our laboratory; anti-DYNC1H1: 1: 100, Santa Cruz Biotechnology, Cat No: sc-7527). After washing three times with PBS, the slides were incubated with secondary antibodies for 1 hr at room temperature. CyTM3 AffiniPure F (ab')2 Fragment Donkey Anti-Rabbit IgG (H+L) (1: 200, Jackson ImmunoResearch, Cat No: 711166152) and DyLight 488 Horse Anti-Mouse IgG Antibody (1: 400, Vector Laboratories, Cat No: DI-2488) are two secondary antibodies used in this experiment. Images were captured using multiple microscopes in separate locations: confocal laser scanning microscopy (Zeiss LSM 700, Virginia Commonwealth University), Nikon DS-Fi2 Eclipse 90i Motorized Upright Fluorescence Microscope (The C.S. Mott Center for Human Growth and Development, Department of Obstetrics & Gynecology, Wayne State University), and Olympus IX-81 microscope equipped with a spinning disc confocal unit (Dr. James G. Granneman's laboratory and Physiology Department, Wayne State University).

## Generation of *Dnali1* cKO mice

*Dnali1<sup>COIN/COIN</sup>* mice were generated at the Center for Mouse Genome Modification at University of Connecticut, and *Stra8-iCre* mice were purchased from the Jackson Laboratory (Stock No: 008208). Cre recombinase was shown to be only active in male germ cells (*Sadate-Ngatchou et al., 2008*). To generate the germ cell-specific *Dnali1* knockout mouse model, 3- to 4-month-old *Stra8-iCre* males were crossed with 3- to 4-month-old *Dnali1<sup>COIN/COIN</sup>* females to obtain *Stra8-iCre; Dnali1<sup>COIN/+</sup>* mice. The 3- to 4-month-old *Stra8-iCre; Dnali1<sup>COIN/+</sup>* males were crossed back with 3- to 4-month-old *Dnali1<sup>COIN/COIN</sup>* females again. The *Stra8-iCre; Dnali1<sup>COIN/COIN</sup>* were considered to be the homozygous cKO mice, and *Stra8-iCre; Dnali1<sup>COIN/+</sup>* mice were used as the controls. Genomic DNA was isolated to genotype the offspring. The following primers were used for genotyping: *Stra8-iCre* forward: 5'-GTGCAAGCTGAACAACAGGA-3'; *Stra8-iCre* reverse: 5'-AGGGACACAGCATTGGAGTC-3'. An 844 bp PCR product is amplified from the COIN allele of the *Dnali1* with the following primers: forward: 5'-CCTGTGGGAAAGCTAACCCAGC-3'(DliScF5), and reverse: 5'- GCTGGGGATGCGGTGGGCTC –3' (BGHpAr). A 129 bp PCR product is amplified from the wild-type allele using the following primers: forward: 5'- GACAGGGATGGAGGTTGGGAG-3', and reverse: 5'-GAATGAGTGGTCAGGCCTCTG-3' . The *Pacrg* mutant mice were purchased from Jackson Laboratory (Stock No: 000567, 27).

## Assessment of male fertility and fecundity

Sexually mature *Dnali1* cKO and control male mice were each mated with a 2- to 4-month-old wild-type female for at least 2 months. The presence of vaginal plugs was checked, and the pregnancy of females was recorded. The number of pups was counted the day after birth. Average litter sizes are presented as the number of total pups born divided by the number of mating cages.

## Sperm parameters

After breeding studies, the male mice were euthanized by cervical dislocation following anesthesia. Sperm were washed out from the cauda epididymis in 37°C PBS. Cells were counted using a hemocytometer chamber under a light microscope, and sperm number was calculated by standard methods. Motility percentages and velocities (average path velocity) were then analyzed using ImageJ (National Institutes of Health, Bethesda, MD, USA) and the plug-in MTrackJ.

## Histology of testis and epididymis

Testes and epididymides of adult mice were collected and fixed in 4% PFA in PBS at 4°C overnight. The tissues were embedded in paraffin, sectioned at 5 μm thickness, deparaffined, and stained with hematoxylin and eosin (H&E) using standard procedures. Some testes were also fixed in Bouin's solution and examined after PAS staining at the Henry-Ford Histology Core. Slides were examined using a BX51 Olympus microscope (Olympus Corp, Melville, NY, USA; Center Valley, PA, USA), and photographs were taken with a ProgRes C14 camera (Jenoptik Laser, Germany).

## Transmission electron microscopy

Mouse testes were fixed by incubation in 3% glutaraldehyde (Grade I; Sigma-Aldrich Co, St. Louis, MO, USA) in 0.1 M phosphate buffer pH 7 for 2 hr at room temperature. The samples were washed twice in PBS and re-suspended in 0.2 M sodium cacodylate buffer. The samples were then post-fixed by incubation with 1% osmium tetroxide (Electron Microscopy Sciences, Hatfield, UK), after which they were dehydrated by immersion in a graded series of alcohol solutions and embedded in Epon resin (Polysciences Inc, Warrington, PA, USA). Semi-thin sections were cut and stained with toluidine blue-Azure II. Ultra-thin sections (90 nm) were cut with a Reichert Ultracut S ultramicrotome (Reichert-Jung AG, Wien, Austria) and were then stained with uranyl acetate and lead citrate. Sections were analyzed with a JEOL 1011 microscope and digital images were acquired with a Gatan Erlangshen CCD camera and Digital Micrograph software.

## Statistical analysis

Statistical analyses were performed using Student's *t* test. *$p < 0.05$ was considered as significant. Graphs were created using Microsoft Excel and GraphPad Prism.

## Acknowledgements

This research was supported by Wayne State University Start-up fund and Wayne State University Research Fund, MCI pilot award (to ZZ), MCI fellowship (to YTY), NIH RO1 HD105944 (ZZ), RO1DK76229 (JGG), and an NIH grant RO 1DK115563 (to DCW). Funding was also obtained from Agence Nationale pour la Recherche (grant FLAGELOME ANR-19-CE17-0014 to AT). We also thank Dr. Scott C Henderson and Frances K White for their assistance with using the confocal microscopy in Microscopy Core Facility of Virginia Commonwealth University together with Alain Schmitt, Jean-Marc Massé, and Azzedine Yacia for TEM analyses in PIME core facility of the Institut Cochin (Paris, France). Tissue H&E staining and PAS staining were conducted at the Henry-Ford Histology Core.

## Additional information

### Funding

| Funder | Grant reference number | Author |
|---|---|---|
| Wayne State University Startup fund | | Zhibing Zhang |
| Wayne State University Research Fund | | Zhibing Zhang |
| Male Contraceptive Initiative fellowship | | Yi Tian Yap |
| Male Contraceptive Initiative pilot award | | Zhibing Zhang |
| National Institute of Child Health and Human Development | | Zhibing Zhang |
| National Institute of Diabetes and Digestive and Kidney Diseases | | James G Granneman David C Williams Jr |
| Agence Nationale pour la Recherche | | Aminata Toure |

The funders had no role in study design, data collection and interpretation, or the decision to submit the work for publication.

### Author contributions

Yi Tian Yap, Data curation, Formal analysis, Validation, Investigation, Methodology, Project administration; Wei Li, Ljljiana Mladenovic-Lucas, Formal analysis, Investigation, Methodology, Project administration; Qian Huang, Investigation, Methodology, Writing - original draft, Project administration; Qi Zhou, David Zhang, Investigation, Methodology, Project administration; Yi Sheng, Siu-Pok Yee, Kyle E Orwig, Methodology, Project administration; James G Granneman, Resources, Formal analysis, Investigation, Methodology, Project administration; David C Williams Jr, Resources, Data curation, Formal analysis, Methodology, Project administration; Rex A Hess, Resources, Data curation, Project administration; Aminata Toure, Formal analysis, Methodology, Project administration; Zhibing Zhang, Conceptualization, Data curation, Supervision, Funding acquisition, Investigation, Writing – review and editing

### Author ORCIDs

Yi Tian Yap  http://orcid.org/0000-0001-6448-2748
Qian Huang  http://orcid.org/0000-0001-6836-5135
Siu-Pok Yee  http://orcid.org/0000-0002-2427-401X
James G Granneman  http://orcid.org/0000-0001-7013-6630
David C Williams Jr,  http://orcid.org/0000-0002-6536-4038
Rex A Hess  http://orcid.org/0000-0003-2649-3563
Zhibing Zhang  http://orcid.org/0000-0002-8615-4478

### Ethics

All animal research was executed in compliance with the guidelines of the Wayne State University Institutional Animal Care with the Program Advisory Committee (Protocol number: 18-02-0534).

### Decision letter and Author response

Decision letter https://doi.org/10.7554/eLife.79620.sa1
Author response https://doi.org/10.7554/eLife.79620.sa2

## Additional files

### Supplementary files

• MDAR checklist

### Data availability

All data generated or analysed during this study are included in the manuscript and supporting file; Source Data files have been provided for Figures 2, 3, 5 and 9.

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
