## [Editor Report]

This study provides new insights into the role of DNALI1, an axonemal dynein component, in the machete, a unique transient structure in elongating spermatids during normal spermiogenesis of male germ cell differentiation. The authors provide convincing evidence that DNALI1 is associated with the MEIG1/PACRG complex, which is part of intra-manchette transport system, and is required for proper flagellar assembly and male fertility.

---

## [Decision Letter]

**Decision letter after peer review:**

Thank you for submitting your article "MEIG1/PACRG associated and non-associated functions of axonemal dynein light intermediate polypeptide 1 (DNALI1) in mammalian spermatogenesis" for consideration by *eLife*. Your article has been reviewed by 3 peer reviewers, one of whom is a member of our Board of Reviewing Editors, and the evaluation has been overseen by Anna Akhmanova as the Senior Editor. The following individual involved in the review of your submission has agreed to reveal their identity: Kazuo Inaba (Reviewer #2).

Essential revisions:

Suggested essential revisions by reviewers in common are as described below. Additional details of reviews from each reviewer to improve and clarify the presentation follows in the separate evaluation section.

1) It is suggested to focus on strengthening the function of PACRG and DNALI1 in intra-manchette transport (IMT) which is particularly interesting and relatively strong. However, this will require a good introduction to IMT vs. IFT, a more generally understood process. It is critical to show PACRG and DNALI1 are in complex from native tissue as all the current data are based on in vitro interaction. To link PACRG function to microtubule-based motor function ofo DNAI1, it will be helpful to show DNAI1 interaction with other axonemal dyneins.

2) Related to 1), it is suggested to tone down other claims unless the authors can add additional experimental support suggested by the reviewers. One particular example of such a claim is sperm individualization. As pointed out by all the reviewers, it can be simply due to sprayed (disintegrated) 9+2 axoneme and the present data do not have the resolution to exclude this possibility. In line with this point, the current title and Figure 10 need a revision.

3) It is necessary to improve the basic characterization of the spermatogenesis phenotype of the conditional knockout of DNAI1.

*Reviewer #1 (Recommendations for the authors):*

Much essential information for the figures and methods that are missing are also addressed. In the current form, unfortunately, this manuscript is not strong to support the current major conclusions. To be reconsidered, it would require a major revision. Below are the specific comments for the manuscript.

1. All the interactions between PACRG and DNALI1 are examined only in in vitro system. Have the authors examined their interaction in the native system – testis? Also, the authors should show the interaction within DNALI1 and MEIG here to show their link in detail (see also point #2 re: Figure 1C)

2. re: Figure 1. First, it is not explained why the authors used CHO cells for the co-localization study and COS1 for the co-IP experiment. In the corresponding text to Figure 1, the authors describe the PACRG localizes at vesicle together with DNALI1 in heterologous expression (Figure 1B). However, the current images do not show vesicle localization. Please show more clear images (magnified or insets, or higher resolution microscopy) to support vesicle localization. Besides, when PACRG and DNALI1 are co-localized, their localization is sided in the opposite direction.

In Figure 1B, the author mentioned "DNALI1-flag was present as a vesicle" in CHO cells, what is this vesicle? Please include the authors' explanation.

In Figure 1C, the same size band is seen in IgG control pull-down, thus it is not clear whether PACRG interaction with DNAI1 is positive. Please provide full details such as lane information. As the interaction is ambiguous due to a positive signal in a negative control group, the specificity of the interaction is not clear. How about testing DNAI1 interaction with MEIG1? If DNAI1 interaction with PACRG/MEIG1 is via direct binding to PACRG, then DNAI1 interaction with MEIG1 should be negative.

3. Re: Figure 2 & Figure 3 – Please consider combining Figure 2 & 3 together as they are complementary to each other in supporting one conclusion. In fact, placing Figure 3 before Figure 2 makes more sense as it would provide the radiational why they co-express those together and run co-purified complex in gel filtration.

For current Figure 2 gel filtration, please add the gel filtration protein profile read by A280 together with the position of markers. What is the size of the co-purified PACRG/DNALI1?

In current Figure 3, Please measure the Pacrg mRNA level in each single or combined expression to support the conclusion. Are they really have the same mRNA level but only the protein level changed? If the stability is really the problem, then please apply MG-132 to inhibit the proteasome activity to support the conclusion.

Most importantly, the authors show DNALI1 stabilizes PACRG expression using in vitro system. However, it is contradictory as PACRG protein levels in the native tissue – testis – in DNALI1 knockout mice are not affected. This discrepancy should be explained. It is unclear what's the biological meaning of PACRG stabilization in the developmental procedure?

4. Re: Figure 4, The authors show the co-localization of the DNALI1 and PACRG at the manchette region only in wild type but placed the immunostaining done in parallel from PACRG-deficient germ in Supplementary Figure 1. These two figures should be combined as Figure 1 for fair comparison and easier view for the readers. Also adding sub-figure labels will be helpful (labeling wt only one time on the top of the figure will do good).

5. Re: Figure 5, please split fertility & litter size graphs as the unit of Y-axis value should be different. Also please show all the individual data points for B-E.

6. Re: Figure 6. Dnali1-knockout sperm show clear defects in sperm tail development. However, it is unclear how many sperm cells have flagellum. The authors should include quantitative information on the presence/absence of the sperm tail to understand the role of Dnali1 in sperm flagellar development.

Based on Figure 6, the authors claim that Dnali1-knockout sperm show a problem in sperm individualization. To fully support this conclusion, the authors should demonstrate that the defect in Dnali1-knockout sperm is not due to axoneme instability (i.e., sprayed 9+2 axoneme) by TEM by showing two 9+2 tails belonging to one sperm head or double head.

7. In histology results (Figure 7), it is unclear how the authors recognize phagocytosis sperm. What's the evidence the pointed germ cell in Figure 7A is undergoing phagocytosis.

8. Re: Figure 9, Please show PACRG localization as well. In Dnali1 cKO elongating spermatid cells, where PACRG is localization? Is it also no longer present in the manchette? This should be included in Figure 9. This is very important as this manuscript only has in vitro data to show PACRG/DNALI1 interaction, but no in vivo data. So far, the author only shows PACRG KO doesn't change DNALI1 manchette localization, DNALI1 cKO doesn't change PACRG protein level in testis. These do not support PACRG interaction with DNALI1. Is a-tubulin staining in Danli1 spermatocytes representative (no to very weak signals compared to other panels).

9. Discussion. In the current Discussion Before page 38, the discussion is either to provide a hypothesis (which should belong to the Introduction to provide the rationale of the study) or repetition/summary of results. From page 38, the content is relevant to Discussion but will be easy if there are subheadings for each section.

10. Re: Figure 10 – The way the author put DNALI1=Motor Complex = IMT is inappropriate, and so do the other three in B, C, and D.

*Reviewer #2 (Recommendations for the authors):*

The authors could rewrite the paper in a better way that is based on the experimental data and their possible interpretation. Intra-manchette transport (IMT) is a phenomenon relatively well recognized by reproductive biologists. However, for a broad range of readers unfamiliar with IMT, the authors would explain more on this, particularly on the difference between IMT and well-known intraflagellar transport (IFT), and on where the authors focus in this paper. Figure 10 is highly speculated and is not unrelated to the data presented in this paper. I am afraid that it gives many misunderstandings to readers, although the author might have added this figure as a working hypothesis. The title "associated and non-associated function" is too obscure to be easily figured out by readers. I can see several careless mistakes in the manuscript. For example, alphabetical labels were repeatedly found in Figure1B panels.

*Reviewer #3 (Recommendations for the authors):*

1. Characterisation of the basic spermatogenesis phenotype in the cKO mouse needs to be improved.

a. The authors need to include body weight and testis weight data.

b. Sperm isolation by cutting the epididymis and allowing sperm to swim out is not appropriate for motility and count analysis. It is dependent on the cuts made as to how many sperm vacate, in addition, non-motile sperm will be less likely to vacate the epididymis. Accurate sperm counts require homogenisation of the epididymis. Similarly, an accurate assessment of sperm motility requires backflushing of the epididymis to collect sperm to prevent biasing for motile sperm.

c. Histological characterisation of the testis needs to be re-done. Testes should be bouin's fixed and stained with Periodic Acid Schiff's stain and hematoxylin (not hematoxylin and eosin) to allow visualisation of the acrosome and accurate staging and proper characterisation of the seminiferous epithelium. A more thorough characterisation of the various stages of spermatogenesis or a larger overview should also then be presented for the cKO vs control testes.

d. The potential spermiation failure needs better characterisation to confirm. Testis weights and testicular daily sperm production should be assessed to determine if the reduction in epididymal sperm counts is due to less sperm being produced in spermatogenesis. If there is no reduction in testis daily sperm production or there is a much greater reduction in epididymal sperm content relative to daily sperm production then this is good evidence for spermiation failure. Stage IX seminiferous tubules should then be used to confirm spermiation failure. The stage XII shown in Figure 7 is not appropriate for confirmation of spermiation failure as by stage XII, nuclear condensation and spermatid elongation is well underway so thin heads could be abnormal step 12 spermatids.

e. Figure 5B what is the is the difference between fertility and litter size. Need to clarify – is fertility pregnancies per plug? The gold standard is to present the average number of pups per plug.

2. The double axoneme phenotype is not commonly seen in KO models and the origins of this might reveal important insight into sperm tail formation. Have the authors characterised the head-to-tail coupling apparatus of these sperm and investigated if this is due to supernumerary basal bodies?

3. The authors should include IF staining of epididymal sperm to show if DNALI1 is localised to the sperm axoneme, and to help clarify if DNALI1 has a direct role in axoneme motility.

4. The origin of the abnormal sperm head shape needs to be defined. The authors suggest it could be due to manchette abnormalities, but they have not actually characterised the manchette. Authors should stain testis sections for α-tubulin to allow stage-specific characterisation of manchette structure, movement, and dissolution from about stages VIII – II-III.

5. The TEM clearly shows abnormalities in sperm tail formation, however, I note that it is from testicular spermatid flagella. Have the authors staged the tubules to assess to ensure that these cross sections are from the end of spermiogenesis i.e. stage VII/VIII? This is important to ensure residual cytoplasm as the differences in the coiling in the mitochondrial sheath and other accessory structures are not just due to tails being at different steps of development.

6. Supplemental Figure 6 – the two KO images are a much earlier step spermatid than the WT (Step 9 vs Step 12-13) and MEIG1 does appear to be at the manchette in the cKO.

7. Paragraph from line 673 needs to be rewritten. It is currently written as if spermiation in mammals follows the exact same events as the sperm individualisation process in *Drosophila*. The processes are different. The bundles of 64 elongated spermatids and individualisation complexes are specific to *Drosophila*. In mammals spermiation is thought to be driven by tubulobulbar complexes and involves the removal of ectoplasmic specialisations – see Wayne Vogl's papers for an overview.

---

## [Author Response]

Reviewer #1 (Recommendations for the authors):Much essential information for the figures and methods that are missing are also addressed. In the current form, unfortunately, this manuscript is not strong to support the current major conclusions. To be reconsidered, it would require a major revision. Below are the specific comments for the manuscript.1. All the interactions between PACRG and DNALI1 are examined only in in vitro system. Have the authors examined their interaction in the native system – testis? Also, the authors should show the interaction within DNALI1 and MEIG here to show their link in detail (see also point #2 re: Figure 1C)

We agree that the interaction between DNALI1 and PACRG should be examined in the testis. We performed CO-IP using a PACRG antibody and found that DNALI1 was pulled-down together with PACRG, this figure is included in the revised manuscript (Figure 4A). However, the size of the endogenous PACRG is about 26 kDa, almost the same size of the IgG light chain (about 25 kDa). We tried multiple protocols to eliminate the IgG bands, but we were not successful. Therefore, we were not able to provide the Western result conducted using the PACRG antibody.

2. re: Figure 1. First, it is not explained why the authors used CHO cells for the co-localization study and COS1 for the co-IP experiment. In the corresponding text to Figure 1, the authors describe the PACRG localizes at vesicle together with DNALI1 in heterologous expression (Figure 1B). However, the current images do not show vesicle localization. Please show more clear images (magnified or insets, or higher resolution microscopy) to support vesicle localization. Besides, when PACRG and DNALI1 are co-localized, their localization is sided in the opposite direction.In Figure 1B, the author mentioned "DNALI1-flag was present as a vesicle" in CHO cells, what is this vesicle? Please include the authors' explanation.In Figure 1C, the same size band is seen in IgG control pull-down, thus it is not clear whether PACRG interaction with DNAI1 is positive. Please provide full details such as lane information. As the interaction is ambiguous due to a positive signal in a negative control group, the specificity of the interaction is not clear. How about testing DNAI1 interaction with MEIG1? If DNAI1 interaction with PACRG/MEIG1 is via direct binding to PACRG, then DNAI1 interaction with MEIG1 should be negative.

CHO cells have clear morphology, and COS^-1^ cells can express exogenous genes at a higher level, that is the reason we have been using the two different cells for different purposes. We explained this issue in the revised manuscript. We agree that “vesicle localization” may not be the most accurate term to describe the observation of DNALI1 and PACRG in the cells and have replaced it with “granule structure” in the revised manuscript. We also noticed that a band that is of similar size is also seen in the IgG control pull-down. This is a common phenomenon in IP experiment. This could be improved by extensively washing. However, too much washing sometimes also reduces the specific signals. Clearly the DNALI1/Flag signal in the MYC antibody pull down lane is stronger than the IgG pull down lane. Interaction between DNALI1 and PACRG is also supported by other experiments, indicting specific interaction between the two proteins. As suggested, we added the IP experiment to examine DNALI1 and MEIG1 showing negative interaction between DNALI1 and MEIG1 in the transfected COS^-1^ cells (Supplemental Figure 1).

3. Re: Figure 2 & Figure 3 – Please consider combining Figure 2 & 3 together as they are complementary to each other in supporting one conclusion. In fact, placing Figure 3 before Figure 2 makes more sense as it would provide the radiational why they co-express those together and run co-purified complex in gel filtration.For current Figure 2 gel filtration, please add the gel filtration protein profile read by A280 together with the position of markers. What is the size of the co-purified PACRG/DNALI1?In current Figure 3, Please measure the Pacrg mRNA level in each single or combined expression to support the conclusion. Are they really have the same mRNA level but only the protein level changed? If the stability is really the problem, then please apply MG-132 to inhibit the proteasome activity to support the conclusion.Most importantly, the authors show DNALI1 stabilizes PACRG expression using in vitro system. However, it is contradictory as PACRG protein levels in the native tissue – testis – in DNALI1 knockout mice are not affected. This discrepancy should be explained. It is unclear what's the biological meaning of PACRG stabilization in the developmental procedure?

As suggested, we’ve reorganized the sequence of figures and have now placed Figure 3 before Figure 2 in the revised manuscript. We apologize for inaccurately identifying the original Figure 2 (now Figure 3) as gel filtration. We’ve updated the manuscript stating that nickel affinity chromatography was performed. The sizes of the two proteins are labeled in the revised figure. We agree that the mRNA level of *Pacrg* should be examined to support the conclusion and have now included the qPCR results in the new Figure 2Ab. We acknowledge that it appears contradictory for PACRG protein levels to be unaffected in the native tissue. This is possibly due to the stabilization of PACRG. In our earlier publication, we showed the result of adding MG132 to the cells, therefore we consider that it is redundant to include this experiment again in this study. PACRG associates with multiple proteins. We have shown that MEIG1 also stabilizes PACRG. in vivo, if DNALI1 is absent, PACRG can also be stabilized by MEIG1, which explains why the PACRG protein level is not changed in the *Dnali1* cKO mice. We discuss this possibility in the revised manuscript.

4. Re: Figure 4, The authors show the co-localization of the DNALI1 and PACRG at the manchette region only in wild type but placed the immunostaining done in parallel from PACRG-deficient germ in Supplementary Figure 1. These two figures should be combined as Figure 1 for fair comparison and easier view for the readers. Also adding sub-figure labels will be helpful (labeling wt only one time on the top of the figure will do good).

We do acknowledge that placing wild-type and PACRG-deficient germ cells in parallel would be easier for the readers. However, placing Supplementary Figure 1 (now Supplemental Figure 2) alongside Figure 4 would not be an equivalent comparison as Figure 4 is looking at isolated germ cells whereas Supplementary Figure 1 is obtained by performing immunostaining on the whole testis section. We agree that labeling wt once on the top of figure is sufficient and have removed the repetitive labels in the revised manuscript.

5. Re: Figure 5, please split fertility & litter size graphs as the unit of Y-axis value should be different. Also please show all the individual data points for B-E.

We recognize the inadequate presentation of Figure 5 and have now updated it to show fertility and litter size on separate figures. Individual data points for all the figures are shown in the revised manuscript.

6. Re: Figure 6. Dnali1-knockout sperm show clear defects in sperm tail development. However, it is unclear how many sperm cells have flagellum. The authors should include quantitative information on the presence/absence of the sperm tail to understand the role of Dnali1 in sperm flagellar development.Based on Figure 6, the authors claim that Dnali1-knockout sperm show a problem in sperm individualization. To fully support this conclusion, the authors should demonstrate that the defect in Dnali1-knockout sperm is not due to axoneme instability (i.e., sprayed 9+2 axoneme) by TEM by showing two 9+2 tails belonging to one sperm head or double head.

All developed sperm had tails, just the morphology was not normal. We appreciate the reviewer’s suggestion to conduct more TEM to show the two 9+2 tails belonging to one sperm head or double head. Our institute does not have a TEM core facility. The samples would have to be sent to our collaborator’s lab in Paris, and the collaborator moved to another institute recently. Due to the complicated situation, and the high cost for the TEM experiment, particularly for the external users, we feel difficult to complete this experiment. However, we conducted other experiment to show abnormal autophagy process in this model. We believe that the new data (supplemental Figure 15 in the revised manuscript) support our conclusion.

7. In histology results (Figure 7), it is unclear how the authors recognize phagocytosis sperm. What's the evidence the pointed germ cell in Figure 7A is undergoing phagocytosis.

As suggested, we revised Figure 7 and provided additional information.

8. Re: Figure 9, Please show PACRG localization as well. In Dnali1 cKO elongating spermatid cells, where PACRG is localization? Is it also no longer present in the manchette? This should be included in Figure 9. This is very important as this manuscript only has in vitro data to show PACRG/DNALI1 interaction, but no in vivo data. So far, the author only shows PACRG KO doesn't change DNALI1 manchette localization, DNALI1 cKO doesn't change PACRG protein level in testis. These do not support PACRG interaction with DNALI1. Is a-tubulin staining in Danli1 spermatocytes representative (no to very weak signals compared to other panels).

This is a very good question and we tried to answer even before we submitted the original manuscript. We have been struggling with poor PACRG antibodies. We have been working on this project for more than 13 years, and some results presented in this manuscript were generated more than 10 years ago. The original polyclonal PACRG antibody was from an investigator at Baylor College of Medicine, and the antibody worked perfectly in both Western blotting and IF. Unfortunately, we ran out of that antibody many years ago. We tried to contact the investigator to request more antibody several times (it appears that the investigator moved to Canada a while ago), but we have never had response. We decided to generated mouse monoclonal PACRG antibodies in our lab about 10 years ago (a company Abmart started new business and provided very preferential price: the company synthesized multiple peptides and made the antibodies with $1500). Several monoclonal PACRG antibodies worked for Western blotting but only one worked for both Western and IF. Given the small volume of monoclonal antibodies, we also ran out of the one that worked for IF. We purchased two commercial PACRG antibodies in preparation for this manuscript, however, these antibodies did not give any specific signal in immunofluorescence stains. We provided the information in the revised manuscript. Given the high cost of purchasing more PACRG antibodies, particularly that another lab is publishing their manuscript, we consider that we cannot wait at the moment. Studying this pathway is a long-term goal of our lab, we will absolutely continue the suggested experiment when appropriate antibody is available.

9. Discussion. In the current Discussion Before page 38, the discussion is either to provide a hypothesis (which should belong to the Introduction to provide the rationale of the study) or repetition/summary of results. From page 38, the content is relevant to Discussion but will be easy if there are subheadings for each section.

We agree that the organization of the Discussion section can be improved. Subheadings for each section of the Discussion have been added in the revised manuscript.

10. Re: Figure 10 – The way the author put DNALI1=Motor Complex = IMT is inappropriate, and so do the other three in B, C, and D.

We agree that the statements in Figure 10 (now Figure 11) are inaccurate. We’ve replaced “=” with “∈” to show that DNALI1 is “part of” the various molecular pathways indicated.

Reviewer #2 (Recommendations for the authors):The authors could rewrite the paper in a better way that is based on the experimental data and their possible interpretation. Intra-manchette transport (IMT) is a phenomenon relatively well recognized by reproductive biologists. However, for a broad range of readers unfamiliar with IMT, the authors would explain more on this, particularly on the difference between IMT and well-known intraflagellar transport (IFT), and on where the authors focus in this paper.

We thank the reviewer for the suggestions to strengthen the manuscript. As suggested, we explained more on the IMT and IFT in the revised manuscript.

Figure 10 is highly speculated and is not unrelated to the data presented in this paper. I am afraid that it gives many misunderstandings to readers, although the author might have added this figure as a working hypothesis.

We did include it as a working hypothesis and we do agree that Fig. 10 (now Fig 11) may be an overstatement. We replaced the “=” sign with “∈” to show that DNALI1 is “part of” the various molecular pathways indicated. We also provided additional data (Supplemental Figure 13 and Supplemental Figure 14) in the revised manuscript.

The title "associated and non-associated function" is too obscure to be easily figured out by readers.

We recognize that the title may be too obscure and substituted with a new title in the revised manuscript.

I can see several careless mistakes in the manuscript. For example, alphabetical labels were repeatedly found in Figure1B panels.

We apologize for the careless mistakes found in the manuscript and have corrected them accordingly.

Reviewer #3 (Recommendations for the authors):1. Characterisation of the basic spermatogenesis phenotype in the cKO mouse needs to be improved.

As suggested, we conducted additional experiments to further characterize the basic spermatogenesis phenotype.

a. The authors need to include body weight and testis weight data.

Body weight and testis weight data are included in the revised manuscript (Figure 5B).

b. Sperm isolation by cutting the epididymis and allowing sperm to swim out is not appropriate for motility and count analysis. It is dependent on the cuts made as to how many sperm vacate, in addition, non-motile sperm will be less likely to vacate the epididymis. Accurate sperm counts require homogenisation of the epididymis. Similarly, an accurate assessment of sperm motility requires backflushing of the epididymis to collect sperm to prevent biasing for motile sperm.

We actually conducted the experiment in the same way the reviewer suggested. The method has been updated in the revised manuscript.

c. Histological characterisation of the testis needs to be re-done. Testes should be bouin's fixed and stained with Periodic Acid Schiff's stain and hematoxylin (not hematoxylin and eosin) to allow visualisation of the acrosome and accurate staging and proper characterisation of the seminiferous epithelium. A more thorough characterisation of the various stages of spermatogenesis or a larger overview should also then be presented for the cKO vs control testes.

As suggested, we conducted additional histologic studies by fixing the testes in Bouin solution and stained the sections by PAS. The new results are now presented in Supplemental Figure 7 and 8.

d. The potential spermiation failure needs better characterisation to confirm. Testis weights and testicular daily sperm production should be assessed to determine if the reduction in epididymal sperm counts is due to less sperm being produced in spermatogenesis. If there is no reduction in testis daily sperm production or there is a much greater reduction in epididymal sperm content relative to daily sperm production then this is good evidence for spermiation failure. Stage IX seminiferous tubules should then be used to confirm spermiation failure. The stage XII shown in Figure 7 is not appropriate for confirmation of spermiation failure as by stage XII, nuclear condensation and spermatid elongation is well underway so thin heads could be abnormal step 12 spermatids.

Because it takes significant time and it is required to breed a large number of mice to have sufficient control and new knockout mice to analyze the testis daily sperm production, we were not able to complete this expensive experiment. However, we further analyzed the Stage IX of the seminiferous tubules are added the result to Figure 7A to further confirm spermiation failure.

e. Figure 5B what is the is the difference between fertility and litter size. Need to clarify – is fertility pregnancies per plug? The gold standard is to present the average number of pups per plug.

As suggested, we revised the Figure 5B (now Figure 5C) in the revised manuscript.

2. The double axoneme phenotype is not commonly seen in KO models and the origins of this might reveal important insight into sperm tail formation. Have the authors characterised the head-to-tail coupling apparatus of these sperm and investigated if this is due to supernumerary basal bodies?

We agree that the double axoneme phenotype should be better characterized. As explained above, it is very difficult to conduct more TEM studies. However, we conducted other experiments to further characterize the abnormal sperm, and the new results are presented as Supp Figure 13 and 15 in the revised manuscript.

3. The authors should include IF staining of epididymal sperm to show if DNALI1 is localised to the sperm axoneme, and to help clarify if DNALI1 has a direct role in axoneme motility.

We agree that it is important to show the localization of DNALI1 in the sperm axoneme. It has been reported that DNALI1 was present in the sperm tail by immunofluorescence staining (reference 22 in the manuscript). Unfortunately, due to time constraint and limitation of core facility available at our institute, we were not able to conduct immune EM to further validate the localization. We will definitely consider the experiment in the future study.

4. The origin of the abnormal sperm head shape needs to be defined. The authors suggest it could be due to manchette abnormalities, but they have not actually characterised the manchette. Authors should stain testis sections for α-tubulin to allow stage-specific characterisation of manchette structure, movement, and dissolution from about stages VIII – II-III.

We agree that stage-specific characterization of manchette structure should be performed to better demonstrate its contribution to the abnormal sperm head shape observed. We’ve performed α-tubulin stain on both testis sections and isolated germ cells and included these in the revised manuscript (Figure 8 and Supp Figure 9).

5. The TEM clearly shows abnormalities in sperm tail formation, however, I note that it is from testicular spermatid flagella. Have the authors staged the tubules to assess to ensure that these cross sections are from the end of spermiogenesis i.e. stage VII/VIII? This is important to ensure residual cytoplasm as the differences in the coiling in the mitochondrial sheath and other accessory structures are not just due to tails being at different steps of development.

We recognize that the current TEM images are not the best representation of abnormalities in sperm tail formation. However, due to limited mice available and lack of TEM core at our institute, it would be very. difficulty for us to conduct more TEM studies.

6. Supplemental Figure 6 – the two KO images are a much earlier step spermatid than the WT (Step 9 vs Step 12-13) and MEIG1 does appear to be at the manchette in the cKO.

We acknowledge the reviewer for pointing out that some MEIG1 is present in the manchette in the *Dnali1* cKO occasionally. One possibility is that the *Dnali1* gene is not 100% disrupted using the cKO strategy; another possibility is that MEIG1 associates with another DNALI1-independent system in the manchette. We discussed the possibilities in the revised manuscript. Due to the limited mice available and low number of the germ cells in the late stage in the *Dnali1* cKO mice, we were not successful to compare the localization of MEIG1 step by step between the control and the *Dnali1* cKO mice. This project is one of the major focus in the lab, and we will absolutely continue the studies in the future.

7. Paragraph from line 673 needs to be rewritten. It is currently written as if spermiation in mammals follows the exact same events as the sperm individualisation process in *Drosophila*. The processes are different. The bundles of 64 elongated spermatids and individualisation complexes are specific to Drosophila. In mammals spermiation is thought to be driven by tubulobulbar complexes and involves the removal of ectoplasmic specialisations – see Wayne Vogl's papers for an overview.

We thank the review for pointing this out and we rewrote this paragraph. We also mentioned the new yeast two-hybrid result using human DNALI1 as the bait. Discovery of the interaction between DNALI1 and γ-actin strongly supports that DNALI1 is involved in spermiation and offers a future study to further dissect the mechanism of DNALI1 in spermiogenesis.